 # HEURIGYM

# AN AGENTIC BENCHMARK FOR LLM-CRAFTED HEURISTICS IN COMBINATORIAL OPTIMIZATION

**Hongzheng Chen**[1*]  **Yingheng Wang**[1*]  **Yaohui Cai**[1*]  **Hins Hu**[1*]  **Jiajie Li**[1*]
**Shirley Huang**[2]  **Chenhui Deng**[3]  **Rongjian Liang**[3]  **Shufeng Kong**[1]
**Haoxing Ren**[3]  **Samitha Samaranayake**[1]  **Carla P. Gomes**[1]  **Zhiru Zhang**[1]

[1] Cornell University    [2] Harvard University    [3] NVIDIA

{hzchen,yingheng}@cs.cornell.edu,
{yc2632,zh223,jl4257}@cornell.edu

## ABSTRACT

While Large Language Models (LLMs) have demonstrated significant advancements in reasoning and agent-based problem-solving, current evaluation methodologies fail to adequately assess their capabilities: existing benchmarks either rely on closed-ended questions prone to saturation and memorization, or subjective comparisons that lack consistency and rigor. In this work, we introduce **HeuriGym**, an agentic framework designed for evaluating heuristic algorithms generated by LLMs for combinatorial optimization problems, characterized by clearly defined objectives and expansive solution spaces. HeuriGym empowers LLMs to propose heuristics, receive evaluative feedback via code execution, and iteratively refine their solutions. We evaluate nine state-of-the-art models on various problems across domains such as computer systems, logistics, and biology, exposing persistent limitations in tool use, planning, and adaptive reasoning. To quantify performance, we propose the Quality-Yield Index (QYI), a metric that captures both solution pass rate and quality. Even top models like GPT-o4-mini-high and Gemini-2.5-Pro attain QYI scores of only 0.6, well below the expert baseline of 1. Our open-source benchmark aims to guide the development of LLMs toward more effective and realistic problem-solving in scientific and engineering domains.

## 1 INTRODUCTION

Recent advancements in Large Language Models (LLMs) have significantly expanded their capabilities in complex reasoning and agent-based problem-solving, enabling applications ranging from automated code generation (Li et al., 2022; Novikov et al., 2025) to dynamic decision-making systems (Schick et al., 2023; Yao et al., 2023). Yet existing evaluation frameworks struggle to rigorously assess the full spectrum of these emergent abilities, often failing to capture the demands of real-world tasks that require iterative reasoning, creative algorithm design, and adaptive tool use. This shortcoming leaves a critical gap in understanding whether LLMs can move beyond pattern recognition to exhibit genuine problem-solving ingenuity in practice.

Current evaluation paradigms fall into two categories with distinct limitations. **(1) Ground-truth-based objective benchmarks** rely on closed-form questions (e.g., multiple-choice mathematics problems) that have become susceptible to rapid performance saturation. Widely used benchmarks such as AIME (Online, 2025), HumanEval (Chen et al., 2021b), and GPQA Diamond (Rein et al., 2024) now exhibit ceiling effects, with state-of-the-art models achieving over 80% accuracy (OpenAI, 2025; Alibaba, 2025; DeepMind, 2025). Even emerging evaluations like Humanity's Last Exam (HLE) (Phan et al., 2025), initially proposed as a rigorous PhD-level test, saw performance leap from 3% to 25% within months of release (OpenAI, 2025). These benchmarks face a dual crisis: their static question banks risk data contamination as models ingest newer training data, while their closed-ended nature fails to reflect real-world problem-solving where solutions are neither unique nor predefined.

---

*Core Contributor

**(2) Judge-preference-based subjective evaluations**, such as Chatbot Arena (Chiang et al., 2024), take a different approach by assessing model quality through pairwise comparisons by humans or LLM-based proxies (Zheng et al., 2023). These benchmarks support a wide range of plausible outputs, making them better suited for open-ended tasks. However, this flexibility introduces high variance: everyday communication tasks are inherently subjective, and judgments often prioritize superficial factors like response structure or emoji usage over substantive reasoning quality (Singh et al., 2025; Zhang et al., 2024). While recent efforts to automate evaluation with LLM-as-a-judge systems show promise, their reliability remains inconsistent across domains (Krumdick et al., 2025), particularly for technical tasks requiring specialized expertise.

To address these limitations, we introduce **HeuriGym**, a new evaluation paradigm with an agentic framework centered on combinatorial optimization problems, which naturally combine *well-defined objectives* with *large solution spaces*. Rather than relying on well-known benchmarks such as SAT or TSP, we assess whether LLMs can produce high-quality solutions to novel yet foundational problems spanning computer systems (Cai et al., 2025; Moffitt & Fegade, 2025), scientific reasoning (Chen et al., 2021a; 2016), computational biology (Dauparas et al., 2025; Wijsman, 2012), logistics (Li & Lim, 2001; Graves et al., 1993), and electronic design automation (Hofmann et al., 2025; Cong & Zhang, 2006). They are ideal for benchmarking LLMs because they resist memorization due to their computational hardness, offer clear metrics for quantitative evaluation, and reflect real-world use cases where optimal solutions are tractable only for small instances. Since no single heuristic dominates across all problems or instances (Wolpert & Macready, 1997), the search space is rich and diverse. Tackling these challenges requires not only algorithmic knowledge but also heuristic reasoning, tradeoff navigation, and creative problem-solving—skills that are still underexplored in current LLM evaluations. Our framework extends beyond conventional static evaluations by implementing an interactive agentic loop: LLMs generate heuristic algorithms, receive execution feedback from a code environment, and iteratively refine their solutions. This process mirrors practical engineering workflows and enables deeper evaluation of multi-step reasoning, tool use, and instruction following.

Our benchmark systematically evaluates LLMs across four dimensions: (1) *tool-augmented reasoning* through integration with external libraries, (2) *multi-step planning* in decomposing complex problems into executable sub-tasks, (3) *instruction fidelity* in adhering to problem constraints, and (4) *iterative refinement* based on runtime feedback. The framework uniquely probes practical creativity—the ability to adapt textbook algorithms or invent novel strategies for large-scale instances where exact methods like integer linear programming (ILP) may fail.

To capture both the number of feasible solutions and their quality relative to expert performance, we introduce a unified metric—the Quality-Yield Index (QYI)—which ranges from 0 (all outputs are incorrect or low-quality) to 1 (expert-level performance). Empirical results reveal substantial performance gap: across nine diverse optimization problems, even state-of-the-art LLMs such as GPT-o4-mini-high (OpenAI, 2025) and Gemini-2.5-Pro (DeepMind, 2025) achieve QYI scores around 0.6, underscoring their limited effectiveness in realistic problem-solving settings. These findings highlight the limitations of current benchmarks, which fail to capture the complex, real-world demands of computational problem-solving—where success requires integrating theoretical understanding, tool proficiency, and adaptive reasoning. The contributions of this work are threefold:

- An open-source benchmark suite of nine combinatorial optimization problems that evaluates LLMs' multi-step reasoning capabilities through realistic programming tasks.

- An end-to-end agentic framework supporting LLM solution generation, automated verification, quantitative evaluation with well-defined metrics, and iterative refinement. The resulting system can also serve as a testbed for exploring more advanced prompting techniques and evolutionary strategies.

- A comprehensive empirical study of cutting-edge LLMs, uncovering their current limitations and offering actionable insights for the development of next-generation models and agents.

## 2 RELATED WORK

**LLMs for Combinatorial Optimization.** Recent LLM-based combinatorial optimization (CO) methods follow two main paradigms. The first emphasizes formalization—translating natural language into structured optimization problems. This direction was initiated by the NL4Opt Competition (Ra-

Table 1: Comparison with other recent benchmarks.

| Subjects | Benchmark | Well-Defined Objective | Large Solution Space | Agentic Setting | Evaluation Metrics |
|---|---|---|---|---|---|
| Frontier Knowledge | Humanity's Last Exam (HLE) (Phan et al., 2025) | ✓ | ✗ | ✗ | Accuracy |
| Software Engineering | HumanEval (Chen et al., 2021b) | ✓ | ✗ | ✗ | PASS@$k$ |
| | BigCodeBench (Zhuo et al., 2025) | ✓ | ✗ | ✗ | PASS@$k$ |
| | LiveCodeBench (Jain et al., 2025) | ✓ | ✗ | ✗ | PASS@$1$ |
| | SWE-Bench (Jimenez et al., 2024) | ✓ | ✗ | ✗ | PASS@$1$ |
| | Commit0 (Zhao et al., 2025) | ✓ | ✗ | ✓ | Pass rate |
| Performance Engineering | KernelBench (Ouyang et al., 2025) | ✗ | ✓ | ✗ | FAST$_p$ |
| Daily-Life Tasks | Chatbot Arena (Chiang et al., 2024) | ✗ | ✓ | ✗ | ELO |
| | $\tau$-Bench (Yao et al., 2025) | ✓ | ✓ | ✓ | PASS$^\wedge k$ |
| Combinatorial Optimization | NPHardEval (Fan et al., 2024) | ✓ | ✗ | ✗ | Accuracy |
| | GraphArena (Tang et al., 2025) | ✓ | ✗ | ✗ | Accuracy |
| | ALE-Bench (Imajuku et al., 2025) | ✓ | ✓ | ✓ | ELO |
| | **HeuriGym (This work)** | ✓ | ✓ | ✓ | SOLVE$_s$@$i$, QYI |

mamonjison et al., 2023), with follow-up work improving domain-specific model training (Xiao et al., 2023; JIANG et al., 2025; Li et al., 2025) and prompting strategies (Yang et al., 2024; AhmadiTeshnizi et al., 2024; Iklassov et al., 2024). While effective on benchmarks, these methods struggle to scale due to their reliance on exact solvers (Gurobi, 2025). The second paradigm focuses on heuristic discovery. FunSearch (Romera-Paredes et al., 2024) and AlphaEvolve (Novikov et al., 2025) use LLMs with evolutionary search to generate novel heuristics, but require evaluating thousands of candidates. Recent approaches (Ye et al., 2024; Liu et al., 2024b; Dat et al., 2025) improve efficiency via meta-heuristic templates, but still limit LLMs to filling in a small portion of the algorithm. LLM4AD (Liu et al., 2024c) offers a platform for evaluating such template-based methods. In contrast, HeuriGym removes reliance on templates or scaffolds. It tasks LLMs with generating complete, self-contained optimization programs, including custom data structures and end-to-end pipelines—better reflecting real-world CO challenges, where success depends on uncovering problem-specific structure and designing bespoke algorithms (Wolpert & Macready, 1997).

**Evaluation on LLMs.** As shown in Table 1, existing LLM benchmarks expose key limitations. Many focus on closed-ended tasks in domains like mathematics (Online, 2025), programming (Chen et al., 2021b; Zhuo et al., 2025; Liu et al., 2023), and specialized knowledge (Rein et al., 2024; Phan et al., 2025; Hendrycks et al., 2021), with fixed ground-truths that are prone to data contamination (see § 1). In contrast, open-ended benchmarks (Chiang et al., 2024; Ouyang et al., 2025) encourage diverse outputs but often lack clear objectives, resulting in inconsistent evaluations. Benchmarks like NPHardEval (Fan et al., 2024) and GraphArena (Tang et al., 2025) assess exact solutions to small NP-hard instances, limiting real-world relevance where heuristic solutions are often preferred for scalability. Our benchmark instead accepts any *feasible* solution that satisfies constraints, enabling broader evaluation of algorithmic reasoning. It tasks LLMs with synthesizing executable code, using external libraries, and refining solutions through execution feedback, mimicking realistic workflows. ALE-Bench (Imajuku et al., 2025) and CO-Bench (Sun et al., 2025) are concurrent efforts that focus on score optimization for classic CO problems using metaheuristics, whereas HeuriGym targets practical, high-impact CO problems from scientific and engineering domains, requiring LLMs to discover problem-specific structure and design tailored heuristics. Unlike ALE-Bench's ELO-style scoring, we propose fine-grained SOLVE$_s$@$i$ and QYI metrics that reveal *which stage* agents fail at and *how close* they are to expert solutions in a multi-round reasoning setting, as detailed in § 3.2.

## 3 HEURIGYM: AN AGENTIC FRAMEWORK FOR HEURISTIC GENERATION

In this section, we introduce our agentic framework for evaluating LLM reasoning via iterative heuristic generation, along with benchmark metrics for quantitative assessment.

### 3.1 OVERVIEW

As illustrated in Fig. 1, our framework presents a formal problem description to the LLM, which is then prompted to generate a complete heuristic algorithm conforming to a standardized function signature.

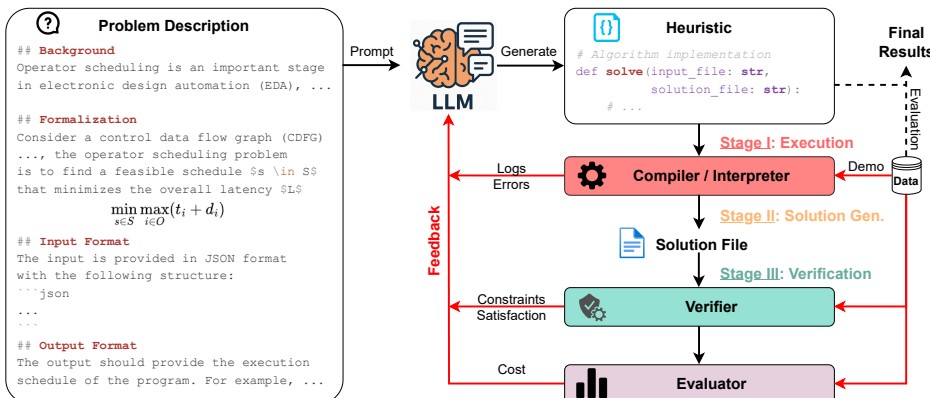

Figure 1: Overview of the HeuriGym agentic framework for heuristic program generation, execution, and verification. We use operator scheduling as an example for the problem description.

It is subsequently compiled (for C++) or interpreted (for Python), and verified for correctness and performance. Crucially, the framework incorporates a feedback loop: execution logs, verification outcomes, and evaluation costs from a small demonstration set are appended back to the prompt, enabling iterative refinement of the LLM-generated solution.

### 3.1.1 PROBLEM DESCRIPTION

As shown on the left of Fig. 1, we use operator scheduling (Cong & Zhang, 2006), a classic optimization problem in electronic design automation, as an example. Each benchmark task is accompanied by a structured problem description with three main parts: **(1) Background**: Introduces the optimization context and key terminology to help the LLM understand the problem setting. **(2) Formalization**: Defines the optimization objective and constraints using mathematical notation (e.g., minimizing latency under hardware resource constraints), guiding the LLM toward objective-oriented algorithm design. **(3) Input/Output Format**: Specifies the structure of input/output files, providing clear expectations for parsing and execution. More details on the problem set can be found in § 4.

### 3.1.2 PROMPT DESIGN

Effective prompt engineering is crucial for leveraging LLMs' capabilities (Wei et al., 2022; Sahoo et al., 2024). We construct both system- and user-level prompts, tailored to each problem. A complete prompt example is provided in Appendix C.

**System prompt.** The system prompt includes machine configuration details (e.g., CPU cores, memory limits), available libraries with version numbers, and task-specific constraints such as execution timeouts. This environment specification instructs the LLM to avoid relying on unrealistic assumptions or producing inefficient solutions that violate runtime limits.

**User prompt.** In the initial iteration, the user prompt includes the problem description and a code skeleton with a predefined function signature. As shown in Fig. 1, the LLM is only provided the interface—function name, input path, and output path—without hints on data structures or algorithmic approache, contrasting with prior work (Romera-Paredes et al., 2024; Liu et al., 2024b; Ye et al., 2024) that often handcrafts partial implementations or restricts the design space. Here, LLMs must reason about the problem holistically: parsing inputs, constructing internal representations, and designing and implementing heuristics from scratch.

### 3.1.3 FEEDBACK LOOP

To emulate a few-shot in-context learning setup (Dong et al., 2024; Liu et al., 2022; Wu et al., 2023), we partition the dataset into a small *demonstration set* (around five instances) and a larger *evaluation set*. Demonstration data is used during the refinement loop to provide timely, example-based feedback to the LLM; the evaluation set is withheld until the model stabilizes its performance.

Each problem includes a domain-specific verifier and evaluator. The verifier ensures constraint satisfaction (e.g., dependency preservation in operator scheduling), while the evaluator calculates the cost based on the given problem objective. If the verifier fails, diagnostic messages are recorded.

After each iteration, we log the LLM-generated solution, execution trace, verification result, and the objective score. These logs are appended to the prompt with the demonstration data in the next iteration, enabling the LLM to learn from past attempts and incrementally improve its output.

## 3.2 METRIC DESIGN

Traditional LLM benchmarks predominantly rely on the PASS@$k$ metric (Chen et al., 2021b; Zhuo et al., 2025; Jimenez et al., 2024), which measures the probability of generating a ground-truth solution within the top-$k$ samples. While PASS@$k$ is effective for single-turn tasks with deterministic ground truths, it falls short in capturing the iterative reasoning and problem-solving abilities required in our multi-round agentic setting. Specifically, it does not reflect whether the LLM can understand problem constraints, debug based on feedback, or iteratively refine its solutions over multiple attempts.

To better evaluate LLMs in this complex setting, we introduce a new metric, denoted as SOLVE$_s$@$i$, which tracks the LLM's ability to solve constrained problems within $i$ iterations:

$$\text{SOLVE}_s@i := \frac{1}{N} \sum_{n=1}^{N} \mathbb{1}\left(\text{pass stage } s \text{ in the } \textit{first } i\text{-th iteration}\right),$$

where $N$ is the total number of instances that are fed to LLMs as inputs, and $s \in \{\text{I, II, III}\}$ denotes the pipeline stage that the solution must pass, each representing a key milestone in agentic reasoning:

- **Stage I: Execution**. The generated program must compile or interpret correctly with all necessary libraries included, and successfully perform basic I/O operations (e.g., reading and writing files).
- **Stage II: Solution Generation**. The program must produce a non-empty output within the predefined timeout and adhere to the expected output format.
- **Stage III: Verification**. The solution must satisfy all problem-specific constraints, as checked by a problem-specific verifier.

However, SOLVE$_s$@$i$ only indicates whether a *feasible* solution is eventually produced through the iterative process—it does not account for solution quality. To address this, we additionally define separate metrics for quality and yield as follows:

$$\text{QUALITY} = \frac{1}{\hat{N}} \sum_{n=1}^{\hat{N}} \min\left(1, \frac{c_n^\star}{c_n}\right) \qquad \text{YIELD} = \frac{\hat{N}}{N},$$

where $c_n$ and $c_n^\star$ represent the cost (i.e., the optimization objective) of the LLM-generated and expert-provided solutions, respectively, and $\hat{N}$ is the number of instances that pass verification (Stage III) in one iteration. We adopt the capped version of quality, which checks whether the LLM matches expert performance (up to a maximum of 1), though an uncapped version can also be used to measure cases where the LLM outperforms the expert. We define a unified metric, the *Quality-Yield Index (QYI)*, as the harmonic mean of quality and yield. This formulation, analogous to the F-score (Van Rijsbergen, 1979), penalizes imbalanced values more strongly than the arithmetic mean:

$$\text{QYI} = \frac{(1 + \beta^2) \cdot \text{QUALITY} \cdot \text{YIELD}}{(\beta^2 \cdot \text{QUALITY}) + \text{YIELD}},$$

where $\beta$ controls the relative importance of YIELD compared to QUALITY. By default, we use $\beta = 1$ in our evaluation. QYI captures both success rate and the relative quality of solutions, enabling holistic evaluation of an LLM's agentic reasoning capabilities, including its capacity for long-horizon planning and iterative refinement. Additionally, we define a weighted QYI by averaging QYI scores with weights proportional to the number of instances in each task, to provide an aggregate measure of overall LLM performance. Nevertheless, we continue to report per-problem QUALITY and YIELD to enable clearer inspection of tradeoffs across individual tasks.

Table 2: Existing combinatorial optimization problems in our HeuriGym benchmark.

| Domain | Problem | References | Difficulty |
|---|---|---|---|
| Electronic Design Automation (EDA) | Operator scheduling | Liu et al. (2024d); Cong & Zhang (2006) | ★ |
| | Technology mapping | Hofmann et al. (2025); Chen & Cong (2004) | ★★ |
| | Global routing | Liang et al. (2024); Liao et al. (2020) | ★★★ |
| Compilers | E-graph extraction | Cai et al. (2025); Willsey et al. (2021) | ★ |
| | Intra-operator parallelism | Moffitt & Fegade (2025); Zheng et al. (2022) | ★★ |
| Computational Biology | Protein sequence design | Dauparas et al. (2025); Kleinberg (1999) | ★ |
| | Mendelian error detection | Lundgren (2025); Sanchez et al. (2008) | ★★ |
| Logistics | Airline crew pairing | Korte & Yorke-Smith (2025); Aggarwal et al. (2023) | ★★ |
| | Pickup and delivery w/ time windows | Taniguchi et al. (2025); Li & Lim (2001) | ★★★ |

## 4  BENCHMARK CONSTRUCTION

This section outlines the construction of our combinatorial optimization benchmark, detailing the principles behind problem selection and providing an overview of the resulting problem set.

### 4.1  PROBLEM SELECTION CRITERIA

Our primary goal is to evaluate an LLM's capacity for reasoning rather than its ability to regurgitate well-known algorithms. To this end, *we intentionally exclude ubiquitous problems* such as TSP (Robinson, 1949) and SAT (Schaefer, 1978)—problems that are so widely studied and frequently included in public datasets that they are likely memorized during pretraining. Instead, we focus on problems that meet the following criteria:

**Limited exposure in the literature.** For each candidate problem, we perform a Google Scholar search and retain it only if the most-cited paper has fewer than 1,000 citations (as of May 2025). This 1,000-citation cutoff is a practical criterion to exclude heavily standardized textbook CO problems that are almost certainly present in LLM training corpora, while still preserving well-defined, peer-reviewed problems. This *empirical* threshold increases the likelihood that the LLM must genuinely reason and design new heuristics rather than rely on cached or widely publicized patterns.

**Clear natural-language specification with well-defined objectives.** Each problem must be clearly expressible using plain language without the need for visual aids. We encode mathematical objectives in LaTeX to eliminate ambiguity, ensuring the LLM receives well-specified instructions.

**Large solution spaces.** We focus on problems that admit vast solution spaces with many feasible outputs, encouraging creative exploration and reasoning rather than narrow pattern recognition (Hughes et al., 2024); in our benchmark, a single instance can present search spaces orders of magnitude larger than those of most existing benchmarks.

**Scalable data instances.** Each problem includes two disjoint sets of instances: a small-scale demonstration set and a large-scale evaluation set, differing by at least an order of magnitude. The demonstration set supports few-shot prompting and iterative refinement, while the evaluation set is reserved for final performance testing, as discussed in § 3.1.3.

**Reproducible expert baselines.** Reference implementations are included in the benchmark repository to ensure fair comparisons in future studies. These expert baselines are often problem-specific heuristic methods, representing the *best-known results* from the literature, and are denoted as QYI 1.0 in our evaluation to highlight the performance gap. If some problems can be expressed as mixed-integer programs, we additionally provide formulations that can be run with commercial solvers like Gurobi. These are just for understanding the gap between heuristic solutions and the optimal solution on small-scale instances.

We prioritize domains with real-world impact, where even small gains yield significant societal or industrial benefits. Many selected problems remain open, with heuristics far from theoretical bounds—offering a compelling testbed for LLMs.

### 4.2  DATASET STATISTICS

The initial release of HeuriGym spans nine optimization problems across scientific and engineering domains, covering fundamentally different types such as covering, scheduling, and routing in Table 2.

Each problem provides five demonstration instances that capture different constraint requirements and provide representative guidance, and dozens of large-scale evaluation instances, totaling 218 (distribution in Appendix G). A small number of demonstration instances is a practical consideration to reduce evaluation time per iteration and also aligns with few-shot LLM usage (Wei et al., 2022; Dong et al., 2024). The effectiveness of the demonstration set is further analyzed in § 5.2.

For *each* instance, the search space is enormously large, growing combinatorially with problem size. This yields many distinct valid solutions rather than a single ground truth, making exhaustive search entirely infeasible. In several tasks, the search space far exceeds even astronomical scales (e.g., $10^{65,000}$ for intra-operator parallelism), and when combined with hard constraints and non-linear objectives, the resulting problems cannot be handled by commercial solvers and are substantially more challenging than typical closed-form optimization benchmarks. All datasets are derived from realistic sources and real-world applications, enhancing the benchmark's practical relevance. Notably, most problems are NP-hard and feature complex constraints, resulting in a compact yet highly challenging problem suite. In addition, we reserve hundreds of instances as private test sets for future release. A detailed description of each problem is provided in Appendix D, and empirical results (§ 5) show that the benchmark remains substantially difficult for state-of-the-art LLMs.

To ensure clarity and correctness, we adopt a human-in-the-loop process for problem specification. A human annotator first drafts the natural-language description, then uses a weaker LLM such as DeepSeek-V3 (Liu et al., 2024a) to highlight any unclear or ambiguous statements. The human and model iteratively resolve discrepancies until the specification is unambiguous and fully aligned with the intended semantics. Importantly, this procedure is solely for refining the problem descriptions but *not* for improving solver performance. The assumption is that if a weaker model can successfully validate the specification's unambiguity and coherence, a stronger model will inherently possess sufficient understanding. The full prompt template used for refining problem descriptions is provided in Appendix C.4.

Each problem includes a task-specific verifier and evaluator to assess solution pass rate and quality. A separate reviewer ensures the expert solver reproduces published results and passes both checks.

Looking forward, we plan to extend HeuriGym along two axes: (1) *breadth*, by incorporating additional combinatorial optimization problems from underexplored scientific domains; and (2) *depth*, by scaling existing problems to larger instance sizes and tighter constraint settings. Community contributions are welcome, provided new problems satisfy the selection criteria articulated above.

## 5 EVALUATION

To evaluate the reasoning capabilities of LLMs on CO problems, we benchmark nine prominent models released in late 2024 and mid-2025 (§ E.1). These models represent the current state-of-the-art in general-purpose LLMs and rank among the top entries on OpenRouter (OpenRouter, 2025) and Chatbot Arena leaderboards (Chiang et al., 2024). We exclude smaller models due to the complexity of the benchmark tasks. All evaluations are conducted via official APIs to ensure reproducibility. We adopt the agentic workflow in Fig. 1, constraining each model to generate Python programs that solve the given problems under fixed resource limits: a maximum of 8 CPU cores and problem-specific timeouts. We also allow the models to access the given Python libraries for external tool use. Full details of the experimental settings and results of each problem can be found in § E. Notice the main goal of these experiments is *not* to present an optimal pipeline, but rather to establish a baseline for current LLM capabilities and to demonstrate that HeuriGym serves as a common testbed on top of which more advanced prompting strategies and sophisticated agentic workflows can be developed. We further discuss these potential improvements in § 6.

### 5.1 OVERALL PERFORMANCE

For the overall evaluation, we fix the generation temperature at 0, following standard practice in recent LLM benchmarks (Ouyang et al., 2025; Yao et al., 2025; Phan et al., 2025). This ensures deterministic outputs and eliminates randomness across runs. Notably, OpenAI's o-series models only support a fixed temperature of 1.0 (OpenAI, 2025). We measure the multi-round performance using the $\textsc{solve}_s @ i$ metric, where $i \in \{1, 5, 10\}$ indicates the number of iterations allowed.

Table 3: Overall $\text{SOLVE}_s@i$ metric of models on the whole HeuriGym benchmark.

| Model | $\text{SOLVE}_{\text{III}}$ @10 | @5 | @1 | $\text{SOLVE}_{\text{II}}$ @10 | @5 | @1 | $\text{SOLVE}_{\text{I}}$ @10 | @5 | @1 |
|---|---|---|---|---|---|---|---|---|---|
| DeepSeek-V3 | 46.8% | 42.7% | 14.2% | 87.6% | 83.0% | 66.1% | **100.0%** | **100.0%** | 90.8% |
| DeepSeek-R1 | 73.4% | **72.9%** | 44.0% | 88.1% | 88.1% | 60.6% | **100.0%** | **100.0%** | 71.6% |
| Gemini-2.5-Flash | 67.4% | 58.3% | 25.2% | 83.9% | 79.4% | 56.4% | **100.0%** | **100.0%** | 72.9% |
| Gemini-2.5-Pro | 65.1% | 64.2% | 20.2% | 89.4% | 89.0% | 42.7% | **100.0%** | **100.0%** | 51.4% |
| LLaMA-4-Maverick | 35.8% | 33.5% | 6.0% | 84.9% | 74.3% | 8.3% | 85.3% | 85.3% | 13.3% |
| LLaMA-3.3 | 33.9% | 33.9% | 20.6% | 78.4% | 78.4% | 40.4% | 99.5% | 99.5% | 61.9% |
| Qwen3-235B | 45.9% | 45.4% | 38.5% | 86.2% | 83.0% | 56.0% | **100.0%** | **100.0%** | 70.6% |
| Claude-3.7-Sonnet | 60.1% | 58.7% | 9.2% | 97.7% | 97.7% | 41.3% | **100.0%** | **100.0%** | 60.1% |
| GPT-o4-mini-high | **74.8%** | 69.7% | **53.2%** | **100.0%** | **100.0%** | **93.1%** | **100.0%** | **100.0%** | **100.0%** |

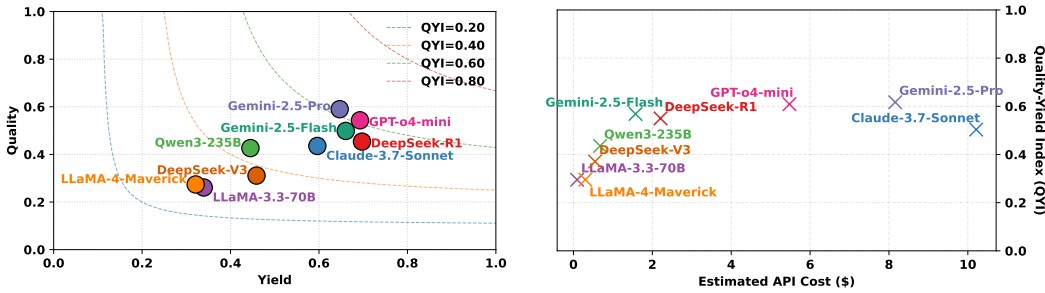

Figure 2: Quality-Yield Index and estimated API cost of different models.

As shown in Table 3, most LLMs fail to solve a large fraction of test cases within a single attempt, as reflected in the $\text{SOLVE}_{\text{III}}@1$ score. Increasing the number of iterations generally improves performance across all models. For instance, the $\text{SOLVE}_{\text{III}}$ success rate rises from 53.2% to 74.8% for GPT-o4-mini-high as $i$ increases, underscoring the importance of iterative refinement in improving LLM-generated solutions. Among all models, GPT-o4-mini-high and DeepSeek-R1 demonstrate high success rates across multiple iterations, highlighting their stronger program repair capabilities.

To assess solution quality, we compare the final LLM-generated programs to expert-designed solutions using the weighted QYI metric defined in § 3.2. As illustrated in Fig. 2, a substantial performance gap remains: even the best-performing model, Gemini-2.5-Pro, achieves a QYI of only 0.62, indicating that its solutions are, on average, just 60% as effective as expert-crafted ones. Several models, such as LLaMA-3.3 and LLaMA-4-Maverick, produce results with QYI scores below 30%, highlighting their limited effectiveness on these tasks. We also estimate the API cost for each model and find that Gemini-2.5-Flash offers the best cost-efficiency relative to its achieved QYI.

We further compare state-of-the-art open-source evolutionary frameworks under the same setting. We fix the outermost evolutionary loop to 10 iterations and use a population size of 10; in total, each method therefore produces hundreds of candidate programs due to multi-step reasoning within each iteration. Further increasing the population size leads to context-length overflows for most of the problems inside HeuriGym. All frameworks use Gemini-2.5-Pro, the best-performing model in our benchmark, as the base LLM. The initial candidate program is generated directly from the problem description. As shown in Table 4, these frameworks perform poorly, often worse than the baseline model. Their main weakness is the lack of incorporating program execution feedback (errors and verification results, § 3.1.3), and their search process breaks context across iterations, causing the system to repeatedly patch the same flawed initial program without real progress. This highlights both the complexity of our benchmark and the need for LLMs to reason more deeply about problem-specific strategies. These frameworks originally only target toy problems under 20 lines of code (e.g., TSP, bin packing), while our benchmark typically requires 300+ lines, making strategy discovery essential rather than relying on prebuilt metaheuristics. These results also underscore the importance of better prompt design and context engineering. Simply appending all sampled programs to the prompt does not scale to our benchmark, limiting the population size per iteration. Moreover, improved mechanisms are needed to consolidate and reconcile feedback from different sampled candidates so that refinements do not conflict with one another.

Table 4: Performance of evolutionary frameworks.

| Frameworks | SOLVE$_{\text{III}}$@10 | QYI |
|---|---|---|
| Gemini-2.5-Pro | 0.6514 | 0.6170 |
| HSEvo (Dat et al., 2025) | 0.5000 | 0.4491 |
| ReEvo (Ye et al., 2024) | 0.4771 | 0.4486 |
| EoH (Liu et al., 2024b) | 0.4954 | 0.4492 |

Table 5: Ablation study on pickup and delivery with time windows.

| # of Demos / # of Feedback Rounds | QYI |
|---|---|
| 5/10 | 0.4196 |
| 3/10 | 0.2829 |
| 0/10 | 0.2351 |
| 5/5 | 0.3330 |
| 5/1 | 0.2350 |

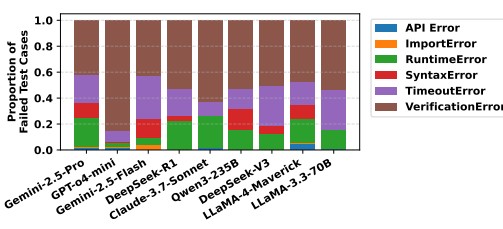

Figure 3: Error classifications.

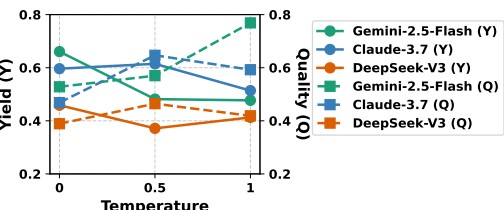

Figure 4: Quality-Yield tradeoff.

To identify common failure modes, we analyze and categorize the most common error types produced by the evaluated models, as shown in Fig. 3. These include: (1) Hallucinated APIs: using nonexistent or outdated library calls. (2) Incorrect algorithmic logic: flawed implementation even when the general approach is reasonable. (3) Constraint misunderstanding: ignoring or misinterpreting problem constraints. (4) Timeouts: no output or the execution time exceeds the given constraints. Additional error cases and examples are listed in Appendix E.8.

## 5.2 ABLATION STUDY

To assess the robustness and sensitivity of LLM performance under different settings, we conduct a set of ablation experiments with full details in Appendix E.

**Temperature.** We evaluate three representative models across the QYI spectrum using temperatures $T \in \{0.0, 0.5, 1.0\}$. Fig. 4 shows that higher $T$ increases diversity and quality but lowers yield due to more invalid outputs (§ E.4). Greedy decoding ($T = 0$) has maximum yield with suboptimal quality, while stochastic sampling ($T = 1$) achieves better quality at the cost of solving fewer problems. This reveals a fundamental trade-off between quality and yield that future LLMs must address.

**Few-shot demonstrations.** We assess the impact of in-context examples by comparing zero-shot, half-shot, and full-shot prompts. Due to budget constraints, these experiments are conducted on a few representative models. Specifically, we evaluate Gemini-2.5-Pro on the pickup and delivery problem—one of the most challenging tasks in our benchmark (full results in § E.5). As shown in Table 5, providing more informative demonstrations significantly boosts the overall performance, especially for tasks involving unfamiliar domains or requiring long-horizon reasoning.

**Feedback rounds.** To evaluate the role of iterative refinement, we vary the number of feedback rounds given to LLMs (1, 5, and 10), keeping the temperature fixed at 0. The results in Table 5 show that later iterations frequently fix logic errors or constraint violations from earlier attempts, underscoring the value of multi-round reasoning. We provide further analysis in § 5.3 and § E.6.

## 5.3 CASE STUDY

We present a case study on technology mapping (Chen & Cong, 2004) to highlight both the promise and current limitations of LLMs, where the task is to cover a logic network with $K$-input lookup tables (LUTs) while minimizing their total count; we fix $K = 6$. As an expert baseline, we use ABC (Berkeley, 2005), a state-of-the-art logic synthesis tool that leverages optimized cut enumeration and dynamic programming (DP)-based covering. We find that top-performing LLMs, such as GPT-o4-mini-high and Gemini-2.5-Pro, can mimic similar heuristic strategies and iteratively refine them

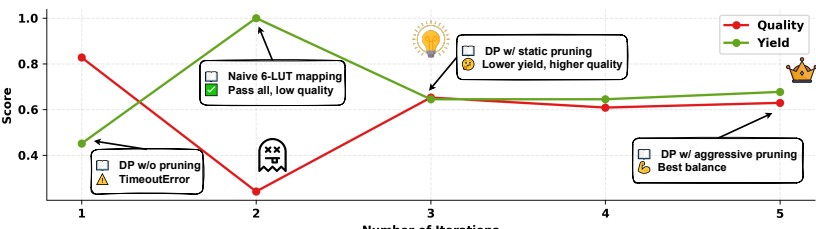

Figure 5: One iterative example of GPT-o4-mini-high on the technology mapping problem.

through feedback. Fig. 5 shows GPT-o4-mini-high explores a range of approaches over multiple iterations, evolving from naive mappings to sophisticated DP-based heuristics with pruning. It finally converges on a strategy that effectively balances yield and quality, achieving the highest QYI. The full generated programs across those iterations are listed in § F.

Nonetheless, a substantial gap remains between LLMs and expert tools (∼60% of expert performance), due to the latter's extensive use of domain-specific optimizations and efficient implementations. This suggests that while LLMs can learn and refine heuristic algorithms, they are not yet capable of generating solutions with expert-level performance in real-world complex optimization tasks.

## 6 DISCUSSION AND LIMITATION

To guide future LLM development, we identify two key challenges: **(1) Correctness**—reducing hallucinations (e.g., incorrect API usage) and improving instruction-following to strictly satisfy problem constraints; and **(2) Performance**—navigating enormous search spaces to discover high-quality solutions. Potential enhancements such as longer context windows and advanced search strategies like iterative best-of-N (§ E.7) may help address these challenges.

While our benchmark and framework offer a promising foundation for evaluating LLMs on combinatorial optimization problems, several limitations remain that suggest directions for future work.

First, all experiments are run in Python, which eases adoption but incurs execution overheads. We report preliminary C++ results in § E.9, but full integration remains challenging due to reliance on domain-specific libraries and the difficulty of generating efficient, correct, and parallel C++ code.

Second, our current agentic pipeline only uses a standard configuration as a baseline and does not incorporate advanced prompting or multi-agent strategies. There are several promising directions for improving context efficiency, such as using summarization (Sahoo et al., 2024) or prompt compression (Chuang et al., 2024). Automatic prompt engineering further expands the design space, with approaches based on gradient descent (Pryzant et al., 2023) or genetic algorithms (Guo et al., 2024) for searching optimal prompts. In addition, multi-agent designs in which different agents handle different components of a decomposed problem have shown strong potential for tackling more complex tasks (Zhou et al., 2025; Gottweis et al., 2025).

Third, the iterative self-refinement process in our agentic workflow can be interpreted as a form of test-time scaling (TTS), analogous to compute-optimal scaling strategies (Snell et al., 2024). This perspective creates opportunities to incorporate techniques such as best-of-N sampling (Stiennon et al., 2020), beam search (Xie et al., 2023), evolutionary algorithms (Novikov et al., 2025; Ye et al., 2024), and external autotuner (van Stein et al., 2025), especially with increased iteration budgets. Furthermore, with a robust verifier in place, our framework provides a natural platform to investigate self-verification capabilities (Kumar et al., 2024), a promising avenue toward greater LLM autonomy.

Finally, our evaluation pipeline currently relies on formally defined and computationally efficient proxy metrics. While these metrics are useful for benchmarking, they may not fully reflect real-world performance. This gap is especially apparent in (1) scientific domains, where solution quality must ultimately be validated through physical experiments, and (2) engineering domains like EDA, where quality must be confirmed through time-consuming backend synthesis. A key challenge for future research lies in bridging this gap while managing the latency introduced by longer feedback loops.

We believe HeuriGym can serve as a shared testbed and foster interdisciplinary collaboration.

## ACKNOWLEDGEMENT

We would like to thank Aaron M. Ferber, Delia Qu, Wenting Zhao, and Cunxi Yu for their constructive feedback on the initial proposal of this project. We are also grateful to Anthony Agnesina and Wen-Hao Liu for providing the initial global routing benchmark, and to Niansong Zhang for generously providing OpenRouter credits. The Cornell authors are supported in part by ACE, one of the seven centers in JUMP 2.0, a Semiconductor Research Corporation (SRC) program sponsored by DARPA; NSF Award #2118709; the AI2050 Senior Fellowship, a Schmidt Sciences program; the National Science Foundation (NSF); the National Institute of Food and Agriculture (USDA/NIFA); and the Air Force Office of Scientific Research (AFOSR).

## ETHICS STATEMENT

In the development of this work, we have been guided by the ICLR Code of Ethics, aiming to contribute positively to the field of combinatorial optimization and human well-being. We acknowledge that, like any new dataset, the content presented could be misused. Our intention for this work is to advance the field of combinatorial optimization for societal benefit, and we encourage the research community to consider and mitigate potential negative consequences in future applications.

We recognize that the large language models utilized may reflect existing biases. By primarily using publicly available models, we aim to be transparent and have avoided using private user data. The use of LLMs is listed in Appendix A. We advocate for continued research into identifying and mitigating biases in LLM-crafted heuristics to ensure fairness and prevent discrimination.

## REPRODUCIBILITY STATEMENT

Our framework, dataset, and benchmark are released under an open-source license and are available in the supplementary material during the review process. All models are accessible via the public API through OpenRouter OpenRouter (2025). The models included in our benchmark are listed in Appendix E.1, with detailed experimental settings provided in Appendix E. To support reproducibility, we also provide step-by-step instructions in our repository.

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

APPENDIX

## A    THE USE OF LARGE LANGUAGE MODELS (LLMS)

In this work, we design an agentic benchmark and evaluate the performance of various LLMs on combinatorial optimization problems. Model specifications are provided in Appendix E.1, and experimental details are described in Appendix E. We ensure that all uses of LLMs are transparent, and our results can be fully reproduced from the submitted supplementary material. Beyond their use in experiments, we also employ LLMs to polish the writing. Importantly, we do *not* use LLMs to propose ideas or design experiments.

## B    UPDATED RESULTS IN FEBRUARY 2026

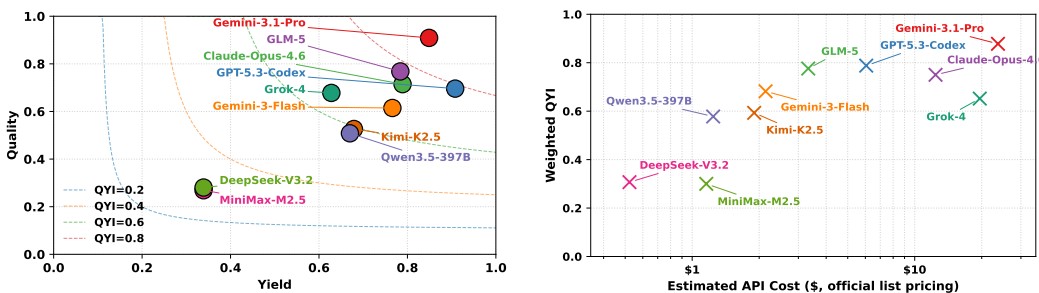

Figure 6: QYI and estimated API cost of the latest ten models in February 2026.

Table 6: Model performance on the latest ten frontier models in February 2026.

| Model | QYI | SOLVE$_{III}$@5 (Yield) | SOLVE$_{II}$@5 | SOLVE$_{I}$@5 | Quality |
|---|---|---|---|---|---|
| Gemini-3.1-Pro | **0.878** | 91.7% | 91.7% | 100.0% | **0.947** |
| GPT-5.3-Codex | 0.788 | **93.6%** | **100.0%** | **100.0%** | 0.773 |
| GLM-5 | 0.776 | 80.3% | 93.6% | **100.0%** | 0.849 |
| Claude-Opus-4.6 | 0.750 | 83.9% | 99.5% | **100.0%** | 0.849 |
| Gemini-3-Flash | 0.682 | 78.0% | 87.6% | **100.0%** | 0.760 |
| Grok-4 | 0.652 | 63.3% | 78.4% | 98.2% | 0.868 |
| Kimi-K2.5 | 0.593 | 71.6% | 88.5% | **100.0%** | 0.730 |
| Qwen3.5-397B | 0.578 | 68.3% | 88.1% | **100.0%** | 0.709 |
| DeepSeek-V3.2 | 0.307 | 33.9% | 84.9% | **100.0%** | 0.626 |
| MiniMax-M2.5 | 0.300 | 33.9% | 62.4% | 83.0% | 0.681 |

We extend our evaluation to a broader set of frontier models released prior to February 2026, as summarized in Table 7. Given the substantial performance improvements observed across models in 2025, we limit each model to five iterative attempts in this experiment to generate alternative solutions. All models are evaluated under the same timeout configuration described in Appendix E.2.

As shown in Table 6 and Fig. 6, Gemini-3.1-Pro leads all ten models with a weighted QYI of 0.878, followed by GPT-5.3-Codex, GLM-5, and Claude-Opus-4.6, forming a top tier separated from the remaining six models. Overall, these models demonstrate substantially stronger performance than those evaluated in May 2025 (Fig. 2), indicating a marked improvement in frontier capability over the past year. The left panel of Fig. 6 shows that these four models cluster in the high-yield, high-quality region, near the QYI=0.8 contour, indicating a balanced tradeoff between coverage and solution performance. In contrast, Gemini-3-Flash, Grok-4, Kimi-K2.5, and Qwen3.5-397B constitute a mid tier, while DeepSeek-V3.2 and Minimax-M2.5 rank last by a substantial margin. The right panel of Fig. 6 further contextualizes these results by incorporating estimated API cost. Although Gemini-3.1-Pro achieves the highest QYI, it is also among the most expensive models, while GPT-5.3-Codex and GLM-5 achieve competitive QYI at moderately lower cost. Notably, GLM-5 lies close to the Pareto frontier in the cost-QYI plane, offering near-top-tier performance at substantially lower price than Gemini-3.1-Pro.

Table 7: Model specifications with API names and official pricing (per million tokens).

| Organization | Model | API Name | Price ($In/$Out) |
|---|---|---|---|
| Google | Gemini-3.1-Pro | `google/gemini-3.1-pro-preview` | 2.00/12.00 |
| OpenAI | GPT-5.3-Codex | `gpt-5.3-codex` | 1.75/14.00 |
| Anthropic | Claude-Opus-4.6 | `claude-opus-4-6` | 5.00/25.00 |
| Zhipu AI | GLM-5 | `z-ai/glm-5` | 1.00/3.20 |
| Google | Gemini-3-Flash | `google/gemini-3-flash-preview` | 0.50/3.00 |
| xAI | Grok-4 | `x-ai/grok-4` | 3.00/15.00 |
| Moonshot AI | Kimi-K2.5 | `moonshotai/kimi-k2.5` | 0.60/3.00 |
| Alibaba | Qwen3.5-397B | `qwen3.5-397b-a17b` | 0.55/3.50 |
| Minimax | Minimax-M2.5 | `minimax/minimax-m2.5` | 0.30/1.20 |
| DeepSeek | DeepSeek-V3.2 | `deepseek-v3.2` | 0.28/0.42 |

The ordering by $\text{SOLVE}_{\text{III}}@5$ partially diverges from weighted QYI: GPT-5.3-Codex achieves the highest coverage at 93.6% (204/218), placing it above Gemini-3.1-Pro (91.7%), yet Gemini-3.1-Pro leads on weighted QYI because its solutions are of substantially higher quality when they are valid. Technology mapping illustrates this tradeoff sharply: GPT-5.3-Codex achieves 100% yield but at a solution quality of only 0.253, whereas Gemini-3.1-Pro covers 94% of cases at quality 0.937. Grok-4 represents the most extreme quality-over-coverage trade-off, achieving the highest average quality among models outside the top four (0.868) but the second-lowest yield (0.628), pulling its weighted QYI to 0.652 despite near-optimal solutions on problems it does solve.

Performance varies widely across problems, revealing a clear difficulty gradient. Operator scheduling, protein sequence design, e-graph extraction, and intra-op parallelism are the most tractable: nine or ten models achieve 100% $\text{SOLVE}_{\text{III}}@5$ on each, with DeepSeek-V3.2 being the sole exception on operator scheduling, where it solves only 1 of 24 cases despite producing syntactically valid code in every attempt. At the opposite extreme, pickup delivery with time windows is the most challenging problem: only Gemini-3.1-Pro achieves full coverage (30/30 valid), six models score 0% despite universally reaching the Stage II, and no other model exceeds 53% coverage. Global routing exhibits a sharp bimodal pattern: Claude-Opus-4.6, GPT-5.3-Codex, and GLM-5 achieve 96–100% $\text{SOLVE}_{\text{III}}@5$, while Kimi-K2.5, DeepSeek-V3.2, Minimax-M2.5, and Gemini-Flash each score 0%, with the failure occurring entirely at the output-format stage.

Failure mode analysis via the $\text{SOLVE}_{\text{I}}@5$-to-$\text{SOLVE}_{\text{III}}@5$ gap reveals qualitatively different breakdown patterns across models: GPT-5.3-Codex loses only 5.2 percentage points on average, indicating that nearly all code it produces satisfies constraints when it runs; DeepSeek-V3.2 loses 68.1 points on average, reflecting a structural tendency to generate solvers that execute correctly but violate problem constraints. Iterative refinement provides the largest gains between $\text{SOLVE}_{\text{III}}@1$ and $\text{SOLVE}_{\text{III}}@5$: Gemini-3.1-Pro's $\text{SOLVE}_{\text{III}}@i$ pass rate rises from 24.8% to 87.6% over the first three iterations, a 63-point gain attributable to feedback-driven correction, with only modest additional improvement to 91.7% by iteration five. In contrast, DeepSeek-V3.2 plateaus after three iterations, suggesting its failures reflect systematic algorithmic deficiencies that iterative prompting alone cannot resolve.

## C  PROMPT DESIGN

In this section, we detail the system and user prompts used by the LLM agent, as well as the auxiliary prompt employed to enhance our problem descriptions.

### C.1  SYSTEM PROMPT

Each iteration of our benchmark begins with a task-agnostic system prompt that instructs the LLM to generate and iteratively refine executable heuristics for combinatorial optimization problems. This system prompt is followed by a task-specific problem statement and an input/output specification. The prompt includes placeholders – highlighted in red – that are dynamically instantiated at runtime for each task. For instance, {NUM_CPU_CORES} represents the CPU core limit for the task (default: 8), and {TIMEOUT} specifies the wall-clock time limit (default: 10 seconds).

**System Prompt**

You are a world-class optimization expert and algorithmic problem solver. Your task is to develop a highly efficient solution to the following optimization problem. Please analyze the problem background, mathematical formulation, and I/O specifications with extreme rigor and attention to detail.

Your mission is to devise and implement the most performant algorithm possible, optimizing for both computational efficiency and solution quality. You should leverage your deep knowledge of algorithms, data structures, and optimization techniques to craft a powerful solution. You have complete freedom in your algorithmic approach. Think systematically and creatively. Your goal is to push the boundaries of what's possible within the computational constraints. Please strictly follow the instructions below.

1. A problem template is provided below. You only need to implement the solve function. Do NOT modify the function signature including the data types of the input arguments. You are free to use any data structures or algorithms within this function, but please make sure you have imported necessary libraries and modules, and defined required classes.

2. The evaluation machine has {NUM_CPU_CORES} CPU cores and sufficient memory to run your program. The time limit for this question is {TIMEOUT} seconds. You are free to implement parallel algorithms where appropriate to maximize performance.

3. The Python version is 3.12. You may use any standard Python libraries and only the following third-party libraries:

   - numpy==2.2.5
   - networkx==3.4.2
   - pandas==2.2.3

4. Your response should consist of a complete implementation of the 'solve' function. Do NOT include any explanations, comments, additional text, or Markdown formatting.

5. You will receive execution feedback after the user runs your program, including runtime metrics and correctness evaluation.

## C.2 USER PROMPT

For each problem, the first iteration begins with the following user prompt, which introduces the task and its objective to the LLM, along with a program template that the model is expected to complete.

**User Prompt**

```
# Problem Information
{PROBLEM DESCRIPTION}

# Program Template
def solve(input_file: str, solution_file: str):
    """
    Solve the optimization problem.

    Please do NOT change the function name and arguments.
    Inputs should be read from input_file
    and outputs should be written to solution_file.
    Input and output formats have been specified in the
    problem statement.
    """
    raise NotImplementedError(
        "This is a placeholder implementation you need to fill in."
    )
```

## C.3 Prompts for Improvement Guidance

Based on the feasibility of the final outputs, we issue one of two improvement prompts in subsequent iterations. If any test cases fail, we provide the following prompt:

---

**Improvement Guidance Case 1**

# Feedback from Previous Iteration (Iteration {iteration-1})
These are the test cases and results from the previous iteration:
## Test Case 1: {test_name}
**Input File:**
{content}
**Result:**
{execution_message}

## Test Case 2: {test_name}
**Input File:**
{content}
**Result:**
{execution_message}

...

# Improvement Guidance
The program failed to produce valid solutions for some test cases. Please fix the following issues:

1. Check for compilation errors or runtime exceptions.
2. Ensure the program handles all edge cases and meets the problem constraints correctly.
3. Verify that the input and output format match the expected format.
4. Make sure all required functions are implemented correctly, and no external forbidden libraries are used.
5. If the program is not able to produce valid solutions for any test case, please try to find the root cause and fix it.
6. If the program is able to produce valid solutions for some test cases, please try to improve the solution.

---

Otherwise, if all test cases pass verification, we issue the following prompt:

---

**Improvement Guidance Case 2**

# Feedback from Previous Iteration (Iteration {iteration-1})
...

# Improvement Guidance
Please carefully observe the problem structure and improve upon this program by:

1. Addressing any weaknesses in the previous approach.
2. Introducing more advanced or efficient algorithms.
3. Focusing on improving performance for test cases.

Your goal is to improve the solution for as many test cases as possible, with special attention to those where the previous solution performed poorly.

---

## C.4 REFINEMENT PROMPT FOR PROBLEM DESCRIPTIONS

To ensure clarity and correctness in problem specification, we employ a human-in-the-loop process. Specifically, we prompt a weaker LLM to flag any unclear or ambiguous statements in the task description. The following prompt is used for this purpose:

---

**Refinement Prompt for Problem Descriptions**

If you were to solve the programming task below, do you have any questions? Is there anything I should clarify before you begin writing code?

# Problem Description
{PROBLEM DESCRIPTION}

---

## C.5 EXAMPLE PROBLEM DESCRIPTION

The following provides an example problem description for operator scheduling. For other problems, please refer to our repository.

```
## Background

High-level synthesis (HLS) is an important stage in electronic design automation (EDA),
↪   aimed at translating a high-level program specification (e.g., written in C/C++ or
↪   SystemC) into a cycle-accurate hardware implementation. After the program is parsed and
↪   analyzed, it is typically transformed into an intermediate representation known as a
↪   Control Data Flow Graph (CDFG). This graph captures the operations (e.g., arithmetic,
↪   memory accesses) and their control/data dependencies. The CDFG can further be processed
↪   into a Directed Acyclic Graph (DAG) to facilitate scheduling and optimization.

One of the core challenges in HLS is operator scheduling, which determines the exact control
↪   step (or cycle) at which each operation is executed, while satisfying data dependencies
↪   and resource constraints. Efficient scheduling plays a critical role in optimizing
↪   design quality in terms of performance, area, and power.

## Formalization

Consider a CDFG with $n$ operation nodes $o_i$, where $i \in O = \{1, 2, \ldots, n\}$, and a
↪   precedence relation $\prec$ on $O$ that captures operation dependencies. Each operation
↪   $o_i$ is associated with a cycle delay $d_i \in \mathbb{Z}^+$ and a resource type $r_i
↪   \in R = \{1, 2, \ldots, k\}$. Let $T = \{0, 1, 2, \ldots, L\}$ represent the set of
↪   control steps (c-steps), and define a schedule as an $n$-tuple $s = (t_1, t_2, \ldots,
↪   t_n)$, where $t_i \in T$ denotes the start time (c-step) of operation $o_i$.

A schedule $s$ is feasible if it satisfies all data dependencies:
$\forall i, j \in O: i \prec j \Rightarrow t_i + d_i \leq t_j$.
Let $S$ denote the set of all feasible schedules. For a given schedule $s$, let $N_r(t)$ be
↪   the number of operations that use resource $r$ in control step $t$, and define the total
↪   usage of resource $r$ as $N_r = \sum_{t \in T} N_r(t)$.

Given a bound $G_r$ on the number of available instances for each resource type $r \in R$,
↪   the operator scheduling problem is to find a feasible schedule $s \in S$ that minimizes
↪   the overall latency $L$, defined as
$\min_{s \in S} \max_{i \in O} (t_i + d_i)$,
subject to the resource constraints
$\forall r \in R, t \in T: N_r(t) \leq G_r$.

## Input Format
The input is provided in JSON format with the following structure:

```json
{
  "name": "input",
  "delay": {
    "mul": 3,
    "sub": 1
  },
  "resource": {
    "mul": 2,
    "sub": 1
  },
  "nodes": [
    ["n1", "mul"],
    ["n2", "mul"],
    ["n3", "sub"]
```

```
  ],
  "edges": [
    ["n1", "n3", "lhs"],
    ["n2", "n3", "rhs"]
  ]
}
```

Where:
- `name`: Name of the input graph
- `delay`: Maps each resource type to its execution delay in cycles
- `resource`: Maps each resource type to the number of available functional units
- `nodes`: List of nodes, where each node is represented as `[node_id, resource_type]`
- `edges`: List of edges, where each edge is represented as `[source_node, target_node,
↪  edge_name]`

## Output Format
The output should provide the execution schedule of the program, indicating the start cycle
↪  of each operation. For example, the following output means that `n1` and `n2` start at
↪  cycle 0, while `n3` starts at cycle 3:
```
n1:0
n2:0
n3:3
```

# D    PROBLEM SET

In this section, we provide more details on the problems included in Table 2. For a representative problem description used in the prompts, please consult our repository for additional details.

## D.1    OPERATOR SCHEDULING

Operator scheduling is a critical stage in high-level synthesis (HLS) (Cong et al., 2011; Pal et al., 2022), the process of converting behavioral hardware descriptions into register-transfer level (RTL) implementations. This task involves carefully assigning each operation to a specific clock cycle while managing a variety of constraints such as data dependencies, resource availability, and performance targets. The effectiveness of the scheduling process is vital, as it directly influences key design metrics including area, power consumption, and execution time, making it an important focus in the field of electronic design automation (EDA).

Over the years, researchers have developed a wide range of techniques to tackle the inherent challenges of operator scheduling in HLS. Exact methods, such as those based on integer linear programming (ILP) (Hwang et al., 1991; Oppermann et al., 2016), can provide optimal solutions but often suffer from scalability issues. As a result, many commercial and academic HLS tools (Xilinx, 2025; Canis et al., 2011) rely on heuristics to achieve practical, near-optimal results. Traditional heuristic approaches, including priority-function-based methods (Shen et al., 2019; Parker et al., 1986; Paulin & Knight, 2002), focus on balancing resource utilization with performance requirements. Notably, methods leveraging systems of difference constraints (SDC) enable an efficient formulation that captures a rich set of scheduling restrictions and casts the optimization objective into a linear programming (LP) framework (Cong & Zhang, 2006; Dai et al., 2018). More recently, the incorporation of machine learning techniques (Chen & Shen, 2019; Liu et al., 2024d) has further advanced the state-of-the-art, enhancing both scheduling efficiency and solution quality in the face of increasingly complex hardware designs.

## D.2    TECHNOLOGY MAPPING

Technology mapping, in the context of logic synthesis for integrated circuits and field-programmable gate arrays (FPGAs), is the process of converting a logic network into an equivalent network of standard cells or logic resources from a specific technology library. The objective is to optimize key design metrics such as area, delay, and power consumption. It is a crucial step in the VLSI design flow and FPGA design flow, determining the actual physical implementation of a design.

Here in our problem setting, we focus on area-optimal technology mapping for lookup table (LUT)-based FPGAs. Given an input logic network, the goal is to cover the network with $K$-input subgraphs,

each of which can be implemented by a $K$-LUT, while minimizing the number of LUTs representing the circuit area.

The most widely adopted approaches are cut-based methods, which operate in two stages: cut enumeration and cut selection. In this approach, all feasible $K$-input cuts—i.e., subgraphs with at most $K$ inputs—are enumerated for each node in the boolean network. Then, a dynamic programming-based selection process chooses one cut per node to construct a full LUT cover of the circuit, optimizing for metrics such as area or delay (Chen & Cong, 2004; Cong & Ding, 1994; Mishchenko et al., 2006). A refinement of this approach is known as priority cut pruning, which retains only a limited set of the most promising cuts per node rather than considering all possible cuts. This significantly improves scalability for large circuits and is widely implemented in tools such as ABC (Berkeley, 2005).

### D.3 GLOBAL ROUTING

The global routing problem addresses the challenge of planning signal paths across a chip after logic placement, determining how a set of nets should traverse the layout to ensure connectivity while reserving space for detailed routing. Rather than producing exact wire geometries, global routing generates abstract paths through routing regions. This step must account for routing congestion, layer limitations, and timing criticality, while managing a growing number of nets in modern designs like Very-Large-Scale Integration (VLSI). The quality of the global routing solution plays a critical role in determining the feasibility and effectiveness of downstream routing stages and can ultimately dictate the success or failure of physical design closure.

The problem has been studied extensively via sequential and ILP-based methods. Maze routing, introduced by Lee (2009), laid the groundwork for sequential approaches, with subsequent improvements such as the work by Soukup (1978). For multi-terminal nets, rectilinear Steiner tree methods were developed (Cong & Madden, 1998). However, sequential routing lacks global coordination and often leads to congestion. ILP-based methods formulate routing as a 0-1 programming, concurrently optimizing over all nets with objectives like wire length and capacity constraints. While exact ILP solvers are computationally intensive, relaxation techniques such as randomized rounding (Carden et al., 1996) and multi-commodity network flow models (Shragowitz & Keel, 1987; Albrecht, 2001) have been employed. Interior-point methods for solving the LP relaxation (Vannelli, 2002; Behjat et al., 2006) have also proven effective for scalable and near-optimal routing.

Hu & Sapatnekar (2001) provided a comprehensive survey of global routing techniques for integrated circuits. Moffitt (2009) revisited the problem, offering a historical perspective and highlighting key open challenges that remain unresolved. More recently, to foster the development of advanced global routing methods, Liang et al. (2024; 2025) introduced an ISPD contest that encourages the use of GPU-based techniques to accelerate global routing.

### D.4 E-GRAPH EXTRACTION

E-graph (Chao & Whitehead, 1978; Nelson & Oppen, 1979) is a data structure that compactly represents a set of expressions. Given an input program and a set of rewrite rules, an e-graph is constructed by applying the rules to the program, generating new expressions, and merging equivalent expressions. It has been widely used to represent and explore the huge number of equivalent program space in tensor graph transformation (Yang et al., 2021; Chen et al., 2024b), sparse linear algebra optimization (Wang et al., 2020), code optimization (Laird et al., 2024; Smith et al., 2024), digital signal processor (DSP) compilation (VanHattum et al., 2021; Thomas & Bornholt, 2024), circuit datapath synthesis (Ustun et al., 2022; Cheng et al., 2024), and floating-point arithmetic (Panchekha et al., 2015).

In an e-graph, all functionally equivalent terms are organized in the same equivalent classes, known as e-classes. Nodes within each e-class that represent values or operators are called e-nodes. E-classes are a partition of e-nodes, where each e-node belongs to exactly one e-class. Dependencies in e-graphs are directed, which point from e-nodes to their children e-classes, indicating the operator (e-node) requires the values (e-nodes) from the child e-classes to compute its value.

In e-graph extraction, an optimized term from an e-graph is extracted after rewrites, based on a user-defined cost model. The goal is to produce a functionally equivalent but improved implementation of

the original input program. The e-graph extraction problem is proven to be NP-hard when common sub-expressions are considered (Stepp, 2011; Zhang, 2023).

Existing e-graph extraction methods include exact methods employing ILP (Cheng et al., 2024; Smith et al., 2024). Recently, there has been significant progress in employing heuristics for e-graph extraction. These include a simple working-list method (Panchekha et al., 2015), a relaxation method utilizing gradient descent (Cai et al., 2025), and a specialized method tailored for sparse e-graphs (Goharshady et al., 2024). The dataset used in evaluation for this work primarily comes from SmoothE (Cai et al., 2025).

### D.5 Intra-Operator Parallelism

Intra-Operator Parallelism (IOPDDL), an emerging challenge introduced in the ASPLOS'25 contest track (Moffitt & Fegade, 2025), addresses the complexities of distributed deep learning. Leading teams in this competition have predominantly employed metaheuristic approaches, distinguished by their unique pre-processing and optimization strategies.

The effective distribution of large machine learning models across multiple hardware accelerators is paramount for achieving desired performance in both training and serving applications (Zheng et al., 2022; Zhao et al., 2023; Shi et al., 2023; Rajbhandari et al., 2020; Lepikhin et al., 2020; Du et al., 2024a; Chen et al., 2024a). This task necessitates sharding the computation graph to minimize communication overhead, a process made intricate by the vast number of operations and tensors involved. Specifically, for a given graph where nodes represent operations with distinct execution strategies (each possessing associated cost and memory usage), an optimal strategy must be chosen for every node. The objective is to minimize the aggregate sum of node and edge costs, without exceeding a strict memory usage constraint across all devices at any point. The inherent diversity in topological and memory characteristics of ML models across varied tasks and modalities renders this problem especially demanding.

### D.6 Protein Sequence Design

Understanding how proteins fold into their native three-dimensional structures (Jumper et al., 2021; Watson et al., 2023) is a central problem in structural biology (Min et al., 2022; Du et al., 2024b), traditionally framed as a forward problem: predicting the structure a given amino acid sequence will adopt (Lin et al., 2023; Wang et al., 2024). In contrast, the protein sequence design or inverse folding problem starts from a fixed target structure and seeks sequences that are likely to fold into it. Many works have shown that this inverse formulation not only offers practical applications in protein engineering but also deepens our understanding of sequence–structure relationships (Drexler, 1981; Yue & Dill, 1995; Shakhnovich & Gutin, 1993; Deutsch & Kurosky, 1996; Sun et al., 1995; Lau & Dill, 1990).

A common modeling approach treats sequence design as a global optimization problem over the space of amino acid sequences. Methods developed by Sun et al. (1995), Shakhnovich & Gutin (1993), and others define a fitness function to select sequences with favorable folding properties. These functions are designed to balance positive design (low free energy in the target structure) with negative design (high energy in competing folds), promoting both thermodynamic stability and structural specificity. More recently, people have been working on multi-state design with more or less general fitness functions (Pokala & Handel, 2005; Ambroggio & Kuhlman, 2006; Allen & Mayo, 2010; Negron & Keating, 2013; Yanover et al., 2007; Hallen & Donald, 2016; Vucinic et al., 2020).

In our benchmark, we focus on the Grand Canonical (GC) model (Sun et al., 1995) of protein sequence design. The GC model operates on (i) a detailed three-dimensional geometric representation of a target structure with $n$ residues, (ii) a simplified binary alphabet distinguishing only hydrophobic (H) and polar (P) residues, and (iii) a fitness function $\Phi$ that favors sequences with densely packed hydrophobic cores while penalizing solvent-exposed hydrophobic residues. Despite its simplicity, the H/P model has been shown to capture key qualitative features of real protein structures (Dill et al., 1995; Kamtekar et al., 1993). Several studies (Micheletti et al., 1999; Banavar et al., 1998) have explored the correspondence between sequences optimized under the GC model and those observed in natural proteins. However, a key obstacle has remained: computing an optimal sequence for a given structure is computationally challenging. The brute-force enumeration over all $2^n$ H/P

sequences is infeasible for realistic protein sizes, and the algorithmic complexity of the problem was explicitly raised as an open question by Hart (1997). An efficient algorithm that constructs an optimal sequence in polynomial runtime was introduced later (Kleinberg, 1999) using network flow.

### D.7 MENDELIAN ERROR DETECTION

Chromosomes encode an individual's genetic information, with each gene occupying a specific position known as a locus. At each locus, a diploid organism carries two alleles—one inherited from each parent—forming its genotype. When direct genotyping is not available, researchers rely on the observable traits or phenotypes, which represent sets of compatible genotypes. A group of related individuals, along with their phenotypes at a locus, is organized into a pedigree, where each individual is either a founder or has parents defined within the structure.

Due to experimental and human errors, pedigree data may contain inaccuracies. These errors are classified as either parental errors (incorrect parentage, which we assume do not occur here) or phenotype errors, which can lead to Mendelian errors. A Mendelian error arises when all genotype combinations compatible with observed phenotypes violate Mendel's law that each individual inherits one allele from each parent. Detecting such inconsistencies is computationally challenging; the number of possible genotype combinations grows exponentially with pedigree size, making full enumeration impractical. In fact, verifying consistency has been shown to be NP-complete (Aceto et al., 2004).

Error detection and correction are crucial for downstream tasks like genetic mapping or disease gene localization. However, existing tools are often limited by scalability issues, strong assumptions, or incomplete analysis. To address these limitations, a soft constraint network framework for detecting Mendelian inconsistencies was proposed (Sanchez et al., 2008), estimating the minimum number of required corrections, and suggesting optimal modifications. These problems naturally align with weighted constraint satisfaction and provide a rich testbed for scalable and flexible inference in large, complex pedigrees.

### D.8 AIRLINE CREW PAIRING

The airline crew pairing problem is a well-established topic in operations research. It involves constructing sequences of flight legs—known as pairings—that begin and end at a crew base, cover all scheduled flights, and satisfy a variety of regulatory and contractual constraints. The primary goal is to minimize total crew-related costs, such as wages, hotel accommodations, and deadhead travel, while ensuring legality and operational feasibility. This problem is typically formulated as a set partitioning model and addressed using column generation and branch-and-price techniques (Desaulniers et al., 2006; Kasirzadeh et al., 2017). Foundational systems developed for carriers like American Airlines demonstrated the effectiveness of these methods at scale (Anbil et al., 1992). More recent innovations include dynamic constraint aggregation (Elhallaoui et al., 2005) and machine learning-based pairing generation (Yaakoubi et al., 2020), which are now integral to commercial solvers such as Jeppesen (Jeppesen, 2021) and Sabre (Sabre, 2020), capable of processing monthly schedules with tens of thousands of flights.

In addition to exact methods, heuristic and metaheuristic techniques – such as genetic algorithms, simulated annealing, and local search – have been explored to improve scalability and reduce computation time, particularly for medium-sized instances or disruption recovery (Lučić & Teodorović, 2007; Souai & Teghem, 2009). These hybrid approaches aim to complement exact optimization methods by leveraging historical data and incorporating planner preferences, offering more flexible and adaptive solutions in practice.

### D.9 PICKUP AND DELIVERY PROBLEM WITH TIME WINDOWS

The Pick-up and Delivery Problem with Time Windows (PDPTW), originally proposed by Dumas et al. (1991), is generalized from a classical NP-hard combinatorial optimization problem—the Capacitated Vehicle Routing Problem (CVRP). It introduces additional complexity through precedence constraints, requiring pick-up locations to precede corresponding drop-off locations, and service time windows at each location. The problem can be seen in many logistic and public transportation systems, with the primary objective of minimizing the total travel cost.

Over the past three decades, a wide range of models and algorithms have been proposed to address the PDPTW, with most falling into the category of heuristic or metaheuristic approaches. Prominent works include simulated annealing (Li & Lim, 2001; Bent & Van Hentenryck, 2006), large neighborhood search (Curtois et al., 2018; Ropke & Pisinger, 2006), and iterated local search (Sartori & Buriol, 2020). In contrast, research into exact solution methods has been relatively limited, with the most effective approaches relying on the set partitioning formulation combined with the branch-cut-and-price algorithm (Ropke & Cordeau, 2009; Baldacci et al., 2011). Ropke et al. (2007) provided a comprehensive survey of PDPTW solvers developed up to 2007. Ho et al. (2018) later reviewed more recent advancements up to 2018, with a particular emphasis on PDPTW variants for people transportation, referred to as the Dial-a-Ride problem. Taniguchi et al. (2025) develop a mathematical model and applies heuristics (Genetic Algorithm, Simulated Annealing, Tabu Search) to analyze how time-window constraints affect urban pickup/delivery truck routing and scheduling.

To support algorithm development, several benchmark datasets have been created and maintained. The Li and Lim dataset (Li & Lim, 2001) is widely used and includes instances ranging from 100 to 1000 locations. More recently, Sartori & Buriol (2020) released a larger-scale dataset generated from real-world spatial-temporal distributions.

## E    ADDITIONAL EXPERIMENTS

In this section, we provide more experimental results and analysis on our benchmark.

### E.1    MODELS

The LLMs used in our experiments are listed in Table 5. All models were accessed via official APIs provided by their respective organizations, except for the Meta models, which are accessed through the OpenRouter (OpenRouter, 2025) API.

Table 8: Model specifications with API names and official pricing.

| Organization | Model | API Name | Price ($In/$Out) | Type |
|---|---|---|---|---|
| OpenAI | GPT-o4-mini-high | o4-mini:high | 1.1/4.4 | Reasoning |
| Anthropic | Claude-3.7-Sonnet | claude-3-7-sonnet-20250219 | 3/15 | Reasoning |
| DeepSeek | DeepSeek-V3 | deepseek-chat(0324) | 0.27/1.10 | Base |
| DeepSeek | DeepSeek-R1 | deepseek-reasoner | 0.55/2.19 | Reasoning |
| Google | Gemini-2.5-Flash | gemini-2.5-flash-preview-04-17 | 0.15/3.5 | Reasoning |
| Google | Gemini-2.5-Pro | gemini-2.5-pro-preview-05-06 | 1.25/10.0 | Reasoning |
| Meta | LLaMA-3.3 | meta-llama/Llama-3.3-70B-Instruct | 0.07/0.33 | Base |
| Meta | LLaMA-4-Maverick | meta-llama/Llama-4-Maverick-17B-128E-Instruct | 0.27/0.85 | Base |
| Alibaba | Qwen3-235B | qwen3-235b-a22b | 0.29/2.86 | Reasoning |

### E.2    EXPERIMENTAL SETTINGS

By default, we constrain LLMs to generate Python code for each problem and execute the code on a CPU server, with each instance allocated 8 CPU cores. The timeout for each problem is specified in Table 9.

Table 9: Timeout for each problem.

| Problem | Timeout (sec) |
|---|---|
| Operator scheduling | 10 |
| Technology mapping | 10 |
| Global routing | 300 |
| E-graph extraction | 10 |
| Intra-op parallelism | 60 |
| Protein sequence design | 10 |
| Mendelian error detection | 10 |
| Airline crew pairing | 10 |
| Pickup and delivery w/ time windows | 60 |

### E.3 DETAILED RESULTS ON EACH PROBLEM

We provide the detailed $\text{SOLVE}_s@i$ values for each problem in Tables 10 through 18. The variation in $\text{SOLVE}_s@i$ across different problems highlights the diverse levels of difficulty, as summarized in Table 2. For instance, the global routing problem remains unsolved by all evaluated LLMs – even for generating a single feasible solution. In the case of the pickup and delivery problem, the low $\text{SOLVE}_{\text{III}}@10$ ratio also indicates that current LLMs struggle to consistently satisfy the problem's constraints.

Table 10: $\text{SOLVE}_s@i$ results on operator scheduling problem.

| Model | SOLVE_III @10 | @5 | @1 | SOLVE_II @10 | @5 | @1 | SOLVE_I @10 | @5 | @1 |
|---|---|---|---|---|---|---|---|---|---|
| DeepSeek-V3 | 100.0% | 100.0% | 4.2% | 100.0% | 100.0% | 100.0% | 100.0% | 100.0% | 100.0% |
| DeepSeek-R1 | 100.0% | 100.0% | 100.0% | 100.0% | 100.0% | 100.0% | 100.0% | 100.0% | 100.0% |
| Gemini-2.5-Flash | 100.0% | 100.0% | 0.0% | 100.0% | 100.0% | 0.0% | 100.0% | 100.0% | 0.0% |
| Gemini-2.5-Pro | 100.0% | 100.0% | 100.0% | 100.0% | 100.0% | 100.0% | 100.0% | 100.0% | 100.0% |
| LLaMA-4-Maverick | 20.8% | 0.0% | 0.0% | 100.0% | 4.2% | 0.0% | 100.0% | 100.0% | 4.2% |
| LLaMA-3.3 | 100.0% | 100.0% | 100.0% | 100.0% | 100.0% | 100.0% | 100.0% | 100.0% | 100.0% |
| Qwen3-235B | 100.0% | 100.0% | 100.0% | 100.0% | 100.0% | 100.0% | 100.0% | 100.0% | 100.0% |
| Claude-3.7-Sonnet | 100.0% | 100.0% | 0.0% | 100.0% | 100.0% | 0.0% | 100.0% | 100.0% | 0.0% |
| GPT-o4-mini-high | 100.0% | 100.0% | 100.0% | 100.0% | 100.0% | 100.0% | 100.0% | 100.0% | 100.0% |

Table 11: $\text{SOLVE}_s@i$ results on technology mapping problem.

| Model | SOLVE_III @10 | @5 | @1 | SOLVE_II @10 | @5 | @1 | SOLVE_I @10 | @5 | @1 |
|---|---|---|---|---|---|---|---|---|---|
| DeepSeek-V3 | 0.0% | 0.0% | 0.0% | 100.0% | 100.0% | 100.0% | 100.0% | 100.0% | 100.0% |
| DeepSeek-R1 | 87.1% | 87.1% | 77.4% | 100.0% | 100.0% | 100.0% | 100.0% | 100.0% | 100.0% |
| Gemini-2.5-Flash | 0.0% | 0.0% | 0.0% | 93.5% | 77.4% | 67.7% | 100.0% | 100.0% | 100.0% |
| Gemini-2.5-Pro | 74.2% | 74.2% | 0.0% | 100.0% | 100.0% | 0.0% | 100.0% | 100.0% | 0.0% |
| LLaMA-4-Maverick | 0.0% | 0.0% | 0.0% | 100.0% | 100.0% | 0.0% | 100.0% | 100.0% | 0.0% |
| LLaMA-3.3 | 0.0% | 0.0% | 0.0% | 100.0% | 100.0% | 0.0% | 100.0% | 100.0% | 6.5% |
| Qwen3-235B | 0.0% | 0.0% | 0.0% | 100.0% | 87.1% | 0.0% | 100.0% | 100.0% | 3.2% |
| Claude-3.7-Sonnet | 87.1% | 87.1% | 0.0% | 100.0% | 100.0% | 64.5% | 100.0% | 100.0% | 100.0% |
| GPT-o4-mini-high | 100.0% | 100.0% | 45.2% | 100.0% | 100.0% | 51.6% | 100.0% | 100.0% | 100.0% |

Table 12: $\text{SOLVE}_s@i$ results on global routing problem.

| Model | SOLVE_III @10 | @5 | @1 | SOLVE_II @10 | @5 | @1 | SOLVE_I @10 | @5 | @1 |
|---|---|---|---|---|---|---|---|---|---|
| DeepSeek-V3 | 0.0% | 0.0% | 0.0% | 33.3% | 33.3% | 0.0% | 100.0% | 100.0% | 100.0% |
| DeepSeek-R1 | 0.0% | 0.0% | 0.0% | 0.0% | 0.0% | 0.0% | 100.0% | 100.0% | 100.0% |
| Gemini-2.5-Flash | 0.0% | 0.0% | 0.0% | 20.8% | 0.0% | 0.0% | 100.0% | 100.0% | 100.0% |
| Gemini-2.5-Pro | 0.0% | 0.0% | 0.0% | 100.0% | 100.0% | 0.0% | 100.0% | 100.0% | 0.0% |
| LLaMA-4-Maverick | 0.0% | 0.0% | 0.0% | 0.0% | 0.0% | 0.0% | 0.0% | 0.0% | 0.0% |
| LLaMA-3.3 | 0.0% | 0.0% | 0.0% | 0.0% | 0.0% | 0.0% | 100.0% | 100.0% | 4.2% |
| Qwen3-235B | 0.0% | 0.0% | 0.0% | 0.0% | 0.0% | 0.0% | 100.0% | 100.0% | 100.0% |
| Claude-3.7-Sonnet | 0.0% | 0.0% | 0.0% | 100.0% | 100.0% | 0.0% | 100.0% | 100.0% | 0.0% |
| GPT-o4-mini-high | 0.0% | 0.0% | 0.0% | 100.0% | 100.0% | 100.0% | 100.0% | 100.0% | 100.0% |

Table 13: $\text{SOLVE}_s@i$ results on e-graph extraction problem.

| Model | SOLVE_III @10 | @5 | @1 | SOLVE_II @10 | @5 | @1 | SOLVE_I @10 | @5 | @1 |
|---|---|---|---|---|---|---|---|---|---|
| DeepSeek-V3 | 4.3% | 0.0% | 0.0% | 100.0% | 100.0% | 82.6% | 100.0% | 100.0% | 100.0% |
| DeepSeek-R1 | 100.0% | 100.0% | 100.0% | 100.0% | 100.0% | 100.0% | 100.0% | 100.0% | 100.0% |
| Gemini-2.5-Flash | 100.0% | 100.0% | 0.0% | 100.0% | 100.0% | 0.0% | 100.0% | 100.0% | 0.0% |
| Gemini-2.5-Pro | 100.0% | 100.0% | 0.0% | 100.0% | 100.0% | 100.0% | 100.0% | 100.0% | 100.0% |
| LLaMA-4-Maverick | 0.0% | 0.0% | 0.0% | 100.0% | 100.0% | 0.0% | 100.0% | 100.0% | 0.0% |
| LLaMA-3.3 | 39.1% | 39.1% | 0.0% | 100.0% | 100.0% | 100.0% | 100.0% | 100.0% | 100.0% |
| Qwen3-235B | 87.0% | 87.0% | 87.0% | 100.0% | 100.0% | 100.0% | 100.0% | 100.0% | 100.0% |
| Claude-3.7-Sonnet | 39.1% | 39.1% | 0.0% | 100.0% | 100.0% | 100.0% | 100.0% | 100.0% | 100.0% |
| GPT-o4-mini-high | 100.0% | 100.0% | 39.1% | 100.0% | 100.0% | 100.0% | 100.0% | 100.0% | 100.0% |

Table 14: SOLVE$_s$@$i$ results on intra-op parallelism problem.

| Model | SOLVE$_{III}$ @10 | @5 | @1 | SOLVE$_{II}$ @10 | @5 | @1 | SOLVE$_{I}$ @10 | @5 | @1 |
|---|---|---|---|---|---|---|---|---|---|
| DeepSeek-V3 | 82.1% | 53.6% | 35.7% | 82.1% | 53.6% | 35.7% | **100.0%** | **100.0%** | **100.0%** |
| DeepSeek-R1 | 92.9% | 92.9% | 35.7% | 92.9% | 92.9% | 35.7% | **100.0%** | **100.0%** | 35.7% |
| Gemini-2.5-Flash | **100.0%** | **100.0%** | **100.0%** | **100.0%** | **100.0%** | **100.0%** | **100.0%** | **100.0%** | **100.0%** |
| Gemini-2.5-Pro | 82.1% | 82.1% | 0.0% | 82.1% | 82.1% | 0.0% | **100.0%** | **100.0%** | 0.0% |
| LLaMA-4-Maverick | 96.4% | 96.4% | 3.6% | **100.0%** | **100.0%** | 3.6% | **100.0%** | **100.0%** | 3.6% |
| LLaMA-3.3 | 75.0% | 75.0% | 3.6% | 82.1% | 82.1% | 3.6% | **100.0%** | **100.0%** | **100.0%** |
| Qwen3-235B | 75.0% | 71.4% | 67.9% | 78.6% | 75.0% | 75.0% | **100.0%** | **100.0%** | **100.0%** |
| Claude-3.7-Sonnet | 82.1% | 82.1% | 71.4% | 82.1% | 82.1% | 78.6% | **100.0%** | **100.0%** | 96.4% |
| GPT-o4-mini-high | **100.0%** | **100.0%** | 92.9% | **100.0%** | **100.0%** | **100.0%** | **100.0%** | **100.0%** | **100.0%** |

Table 15: SOLVE$_s$@$i$ results on protein sequence design problem.

| Model | SOLVE$_{III}$ @10 | @5 | @1 | SOLVE$_{II}$ @10 | @5 | @1 | SOLVE$_{I}$ @10 | @5 | @1 |
|---|---|---|---|---|---|---|---|---|---|
| DeepSeek-V3 | 83.3% | 83.3% | 83.3% | **100.0%** | **100.0%** | **100.0%** | **100.0%** | **100.0%** | **100.0%** |
| DeepSeek-R1 | 87.5% | 87.5% | 0.0% | **100.0%** | **100.0%** | 0.0% | **100.0%** | **100.0%** | 0.0% |
| Gemini-2.5-Flash | 95.8% | **95.8%** | **95.8%** | **100.0%** | **100.0%** | **100.0%** | **100.0%** | **100.0%** | **100.0%** |
| Gemini-2.5-Pro | **100.0%** | 95.8% | 0.0% | **100.0%** | 95.8% | 0.0% | **100.0%** | **100.0%** | 4.2% |
| LLaMA-4-Maverick | 83.3% | 83.3% | 0.0% | 95.8% | 95.8% | 0.0% | **100.0%** | **100.0%** | 4.2% |
| LLaMA-3.3 | 12.5% | 12.5% | 12.5% | 95.8% | 95.8% | 95.8% | 95.8% | 95.8% | 95.8% |
| Qwen3-235B | 87.5% | 87.5% | 87.5% | **100.0%** | **100.0%** | **100.0%** | **100.0%** | **100.0%** | **100.0%** |
| Claude-3.7-Sonnet | 58.3% | 45.8% | 0.0% | **100.0%** | **100.0%** | 0.0% | **100.0%** | **100.0%** | 0.0% |
| GPT-o4-mini-high | 91.7% | 91.7% | 91.7% | **100.0%** | **100.0%** | **100.0%** | **100.0%** | **100.0%** | **100.0%** |

Table 16: SOLVE$_s$@$i$ results on mendelian error detection problem.

| Model | SOLVE$_{III}$ @10 | @5 | @1 | SOLVE$_{II}$ @10 | @5 | @1 | SOLVE$_{I}$ @10 | @5 | @1 |
|---|---|---|---|---|---|---|---|---|---|
| DeepSeek-V3 | **100.0%** | **100.0%** | 0.0% | **100.0%** | **100.0%** | 0.0% | **100.0%** | **100.0%** | 0.0% |
| DeepSeek-R1 | **100.0%** | **100.0%** | 0.0% | **100.0%** | **100.0%** | 0.0% | **100.0%** | **100.0%** | 0.0% |
| Gemini-2.5-Flash | **100.0%** | 10.0% | 10.0% | **100.0%** | **100.0%** | **100.0%** | **100.0%** | **100.0%** | **100.0%** |
| Gemini-2.5-Pro | 80.0% | 80.0% | **80.0%** | 80.0% | 80.0% | 80.0% | **100.0%** | **100.0%** | **100.0%** |
| LLaMA-4-Maverick | 60.0% | 60.0% | 60.0% | 60.0% | 60.0% | 60.0% | 60.0% | 60.0% | 60.0% |
| LLaMA-3.3 | 55.0% | 55.0% | 55.0% | 55.0% | 55.0% | 55.0% | **100.0%** | **100.0%** | **100.0%** |
| Qwen3-235B | 55.0% | 55.0% | 0.0% | **100.0%** | **100.0%** | 0.0% | **100.0%** | **100.0%** | 0.0% |
| Claude-3.7-Sonnet | **100.0%** | **100.0%** | 0.0% | **100.0%** | **100.0%** | **100.0%** | **100.0%** | **100.0%** | **100.0%** |
| GPT-o4-mini-high | **100.0%** | 50.0% | 35.0% | **100.0%** | **100.0%** | **100.0%** | **100.0%** | **100.0%** | **100.0%** |

Table 17: SOLVE$_s$@$i$ results on airline crew pairing problem.

| Model | SOLVE$_{III}$ @10 | @5 | @1 | SOLVE$_{II}$ @10 | @5 | @1 | SOLVE$_{I}$ @10 | @5 | @1 |
|---|---|---|---|---|---|---|---|---|---|
| DeepSeek-V3 | **100.0%** | **100.0%** | 0.0% | **100.0%** | **100.0%** | **100.0%** | **100.0%** | **100.0%** | **100.0%** |
| DeepSeek-R1 | **100.0%** | **100.0%** | **100.0%** | **100.0%** | **100.0%** | **100.0%** | **100.0%** | **100.0%** | **100.0%** |
| Gemini-2.5-Flash | 0.0% | 0.0% | 0.0% | 0.0% | 0.0% | 0.0% | **100.0%** | **100.0%** | 14.3% |
| Gemini-2.5-Pro | 0.0% | 0.0% | 0.0% | 0.0% | 0.0% | 0.0% | **100.0%** | **100.0%** | **100.0%** |
| LLaMA-4-Maverick | **100.0%** | **100.0%** | 0.0% | **100.0%** | **100.0%** | 35.7% | **100.0%** | **100.0%** | **100.0%** |
| LLaMA-3.3 | 42.9% | 42.9% | 42.9% | 42.9% | 42.9% | 42.9% | **100.0%** | **100.0%** | **100.0%** |
| Qwen3-235B | 21.4% | 21.4% | 0.0% | **100.0%** | 85.7% | 0.0% | **100.0%** | **100.0%** | 0.0% |
| Claude-3.7-Sonnet | **100.0%** | **100.0%** | 0.0% | **100.0%** | **100.0%** | 0.0% | **100.0%** | **100.0%** | 0.0% |
| GPT-o4-mini-high | **100.0%** | **100.0%** | **100.0%** | **100.0%** | **100.0%** | **100.0%** | **100.0%** | **100.0%** | **100.0%** |

Table 18: SOLVE$_s$@$i$ results on pickup and delivery with time windows problem.

| Model | SOLVE$_{III}$ @10 | @5 | @1 | SOLVE$_{II}$ @10 | @5 | @1 | SOLVE$_{I}$ @10 | @5 | @1 |
|---|---|---|---|---|---|---|---|---|---|
| DeepSeek-V3 | 0.0% | 0.0% | 0.0% | 80.0% | 73.3% | 73.3% | **100.0%** | **100.0%** | **100.0%** |
| DeepSeek-R1 | 16.7% | 13.3% | 3.3% | **100.0%** | **100.0%** | **100.0%** | **100.0%** | **100.0%** | **100.0%** |
| Gemini-2.5-Flash | **96.7%** | **90.0%** | 6.7% | **100.0%** | **100.0%** | **100.0%** | **100.0%** | **100.0%** | **100.0%** |
| Gemini-2.5-Pro | 30.0% | 26.7% | **13.3%** | **100.0%** | **100.0%** | **100.0%** | **100.0%** | **100.0%** | **100.0%** |
| LLaMA-4-Maverick | 0.0% | 0.0% | 0.0% | **100.0%** | **100.0%** | 0.0% | **100.0%** | **100.0%** | 0.0% |
| LLaMA-3.3 | 0.0% | 0.0% | 0.0% | **100.0%** | **100.0%** | 0.0% | **100.0%** | **100.0%** | 0.0% |
| Qwen3-235B | 0.0% | 0.0% | 0.0% | **100.0%** | **100.0%** | **100.0%** | **100.0%** | **100.0%** | **100.0%** |
| Claude-3.7-Sonnet | 0.0% | 0.0% | 0.0% | **100.0%** | **100.0%** | 16.7% | **100.0%** | **100.0%** | **100.0%** |
| GPT-o4-mini-high | 3.3% | 0.0% | 0.0% | **100.0%** | **100.0%** | **100.0%** | **100.0%** | **100.0%** | **100.0%** |

### E.4 ABLATION ON TEMPERATURE

We evaluate various models across different temperature settings, $T \in \{0.0, 0.5, 1.0\}$. For each model, we run 10 iterations per problem and report the highest QYI achieved across these iterations as the final QYI score for that problem. The overall benchmark score is then computed as the arithmetic mean of QYI across all problems. Detailed results are shown in Tables 19 to 21.

In general, improving the temperature can be beneficial to quality as the model becomes more creative, but may harm yield as it may not follow the constraints strictly. Note that yield emphasizes the best iteration that achieves the highest QYI, whereas SOLVE$_{\text{III}}$ reflects the cumulative success rate across iterations; therefore, their values may differ. Additionally, the weighted QYI is not the harmonic mean of weighted yield and weighted quality, as it is computed by aggregating metrics across different problems using a weighted approach.

We also report an uncapped version of the weighted QYI metric[1], which better reflects cases where LLM-generated programs outperform expert solutions on certain test instances. Improvements are underlined in the tables. While this variant achieves slightly higher scores for most models – indicating occasional superior performance – it also confirms that, in the majority of cases, LLMs still lag significantly behind expert solutions.

Table 19: Performance of different models on Temperature $= 0$.

| Model | Weighted Yield | Weighted Quality | Weighted QYI (Capped) | Weighted QYI (Uncapped) |
|---|---|---|---|---|
| Claude-3.7-Sonnet | 0.5963 | 0.4686 | 0.5034 | 0.5034 |
| DeepSeek-R1 | **0.6972** | 0.5775 | 0.5498 | 0.5553 |
| DeepSeek-V3 | 0.4587 | 0.3890 | 0.3707 | 0.3707 |
| Gemini-2.5-Flash | 0.6606 | 0.5281 | 0.5682 | 0.5753 |
| Gemini-2.5-Pro | 0.6468 | **0.6700** | **0.6170** | **0.6228** |
| LLaMA-3.3 | 0.3394 | 0.3521 | 0.2951 | 0.2953 |
| LLaMA-4-Maverick | 0.3211 | 0.3383 | 0.2955 | 0.2955 |
| Qwen3-235B | 0.4450 | 0.4513 | 0.4355 | 0.4423 |

Table 20: Performance of different models on Temperature $= 0.5$.

| Model | Weighted Yield | Weighted Quality | Weighted QYI (Capped) | Weighted QYI (Uncapped) |
|---|---|---|---|---|
| Claude-3.7-Sonnet | **0.6147** | **0.6468** | **0.5437** | **0.5451** |
| DeepSeek-R1 | 0.5138 | 0.5751 | 0.4743 | 0.4812 |
| DeepSeek-V3 | 0.3716 | 0.4645 | 0.3322 | 0.3322 |
| Gemini-2.5-Flash | 0.4817 | 0.5700 | 0.4760 | 0.4828 |
| Gemini-2.5-Pro | 0.4817 | 0.5609 | 0.4767 | 0.4789 |
| LLaMA-3.3 | 0.3991 | 0.4407 | 0.4108 | 0.4108 |
| LLaMA-4-Maverick | 0.3349 | 0.3712 | 0.3050 | 0.3646 |
| Qwen3-235B | 0.4128 | 0.4798 | 0.4269 | 0.4327 |

### E.5 FEW-SHOT DEMONSTRATION

Table 22 highlights the impact of few-shot demonstrations on LLM performance across the entire HeuriGym benchmark. Introducing only a small number of demonstrations (e.g., three) can negatively affect solution quality and success rate, as these examples may not be representative of the overall dataset, leading the model to overfit to them. However, providing a larger set of demonstrations can potentially improve QYI, as the model benefits from greater diversity and can learn more generalizable patterns.

---

[1]The uncapped version of quality is computed as $1/\hat{N} \sum_{n=1}^{\hat{N}} c_n^\star / c_n$, and the uncapped QYI is derived by substituting the original quality metric with this uncapped variant.

Table 21: Performance of different models on Temperature = 1.

| Model | Weighted Yield | Weighted Quality | Weighted QYI (Capped) | Weighted QYI (Uncapped) |
|---|---|---|---|---|
| Claude-3.7-Sonnet | 0.5138 | 0.5924 | 0.4828 | 0.4841 |
| DeepSeek-R1 | 0.5688 | 0.5625 | 0.5313 | 0.5383 |
| DeepSeek-V3 | 0.4128 | 0.4188 | 0.3839 | 0.3841 |
| GPT-o4-mini-high | **0.6927** | 0.6440 | **0.6089** | **0.6158** |
| Gemini-2.5-Flash | 0.4771 | **0.7688** | 0.5030 | 0.5047 |
| Gemini-2.5-Pro | 0.5229 | 0.4893 | 0.4921 | 0.4981 |
| LLaMA-3.3 | 0.3028 | 0.3627 | 0.2868 | 0.2916 |
| LLaMA-4-Maverick | 0.2982 | 0.3271 | 0.2667 | 0.2672 |
| Qwen3-235B | 0.5459 | 0.5228 | 0.5294 | 0.5364 |

Table 22: Impact of few-shot demonstrations on performance (Model: Gemini-2.5-Pro).

| # of Demos | Weighted Yield | Weighted Quality | Weighted QYI |
|---|---|---|---|
| Zero-shot | 0.5872 | 0.7159 | 0.5999 |
| Half-shot | 0.5092 | 0.6526 | 0.5361 |
| Full-shot | 0.6468 | 0.6700 | 0.6170 |

### E.6 FEEDBACK ROUNDS

Table 23 shows that increasing the number of feedback rounds has a nuanced impact on performance. While a moderate number of rounds (e.g., five) can enhance overall quality by guiding the model to refine its solutions, excessive feedback may lead to diminishing returns or even degrade performance. This suggests that too many rounds can overwhelm the model, making it harder to identify and prioritize the most critical information from the feedback.

Table 23: Impact of feedback rounds on performance (Model: Gemini-2.5-Pro).

| # of Feedback Rounds | Weighted Yield | Weighted Quality | Weighted QYI |
|---|---|---|---|
| 1 | 0.6193 | 0.7290 | 0.6253 |
| 5 | 0.6055 | 0.7313 | 0.6259 |
| 10 | 0.6468 | 0.6700 | 0.6170 |

### E.7 ITERATIVE BEST-OF-N SAMPLING

To investigate the benefits of test-time search strategies, we sample $k$ candidate programs in each iteration, evaluate them, and return feedback for all $k$ programs to the LLM. After a fixed number of iterations, we select the best-performing program from the entire pool – a process we refer to as *iterative best-of-N sampling*. The total number of sampled programs is held constant across different values of $k$. This strategy allows the model to explore diverse candidate solutions in parallel and evolve the program based on evaluative feedback.

As shown in Table 24, increasing $k$ leads to better quality of results, indicating that aggregating feedback across multiple candidates allows the LLM to better explore the solution space and improve sampling efficiency by allocating computational budget toward more informative evaluations.

Table 24: Impact of iterative best-of-N sampling on performance (Model: Gemini-2.5-Pro).

| # of Samples @ Iteration | Weighted Yield | Weighted Quality | Weighted QYI |
|---|---|---|---|
| 2@5 | 0.5688 | 0.7698 | 0.6160 |
| 1@10 | 0.6468 | 0.6700 | 0.6170 |

### E.8 ERROR ANALYSIS

In the following, we present representative examples of common errors made by LLMs during heuristic generation. These errors highlight current limitations in code reliability and execution:

- **Import error:** This type of error occurs when the generated code relies on external libraries that are not available in the environment. In the example below, the model attempts to import the `ortools` library, which results in a `ModuleNotFoundError`. Such errors suggest that the model does not strictly follow the instructions given in the prompt.

```
File "operator_scheduling/gemini-2.5-flash-preview-04-17/itera↵
↪    tion4/solver.py", line 2, in <module>
    from ortools.sat.python import cp_model
ModuleNotFoundError: No module named 'ortools'
```

- **API misuse error:** LLMs often misuse APIs due to a misunderstanding of library interfaces. In the following case, the model tries to call `random()` directly from the `random` module, which is not callable.

```
File "intra_op_parallel/o4-mini/iteration3/solver.py", line 64,
↪    in init_jitter
    if len(ci) > 1 and random() < 0.1:
                      ^^^^^^^^
TypeError: 'module' object is not callable
```

- **Syntax error:** Syntax errors are common when the model fails to adhere to basic language rules. In this example, there is an unmatched parenthesis in a `while` loop condition, leading to a `SyntaxError`. Such mistakes typically indicate a lack of code completion validation in the generation process.

```
File "crew_pairing/deepseek-chat/iteration7/solver.py", line 60
    while len(used_legs) < len(df)):
                                  ^
SyntaxError: unmatched ')'
```

- **Runtime error:** Even syntactically and semantically correct code can fail at runtime. In this case, the model modifies a dictionary while iterating over it, which raises a `RuntimeError`. This highlights the model's difficulty in reasoning about the actual executable code in a long context.

```
File
↪    "technology_mapping/llama-4-maverick/iteration2/solver.py",
↪    line 104, in technology_mapping
    for successor in G.successors(node):
RuntimeError: dictionary changed size during iteration
```

### E.9   C++ EXAMPLE

We conduct preliminary experiments on the technology mapping problem by modifying the prompt to instruct the LLM to generate a C++ solution, using the provided function template: `void solve(const std::string& input_file, const std::string& output_file)`.

Integrating C++ into our agentic feedback loop remains challenging due to dependencies on domain-specific libraries and the complexity of parallel execution. As a result, our preliminary experiment with C++ involves only a single iteration of prompting.

Table 25 presents a performance comparison between the Python solution with 10 iterations and the C++ solution with just one iteration. Although the C++ solution does not produce high-quality output in its initial attempt, it already achieves a better yield than the Python solution after 10 iterations – an unexpectedly strong outcome. Notably, the Python solution fails to generate any valid result in its first iteration. This is attributed to the significantly faster execution speed of C++ code, which enables it to avoid the timeout errors frequently encountered by Python in this task.

We expect to see further performance improvement with C++ after we integrate it into the feedback loop in our framework.

Table 25: Impact of C++ code on technology mapping performance (Model: `Gemini-2.5-pro`).

| Language | # of Iterations | Yield | Quality | QYI |
|---|---|---|---|---|
| Python | 10 | 0.7419 | 0.6423 | 0.6885 |
| C++ | 1 | 0.7742 | 0.3493 | 0.4814 |

### E.10  TOKEN USAGE

Table 26 presents an example of token usage when running the complete HeuriGym benchmark across different models. Among them, Gemini-2.5-Pro consumes the most tokens for prompt and completion.

Table 26: Token counts from a single run of HeuriGym across different models.

| Model | Prompt Tokens | Completion Tokens |
|---|---|---|
| Claude-3.7-Sonnet | 1911947 | 298369 |
| DeepSeek-R1 | 1215661 | 702746 |
| DeepSeek-V3 | 1536084 | 122058 |
| GPT-o4-mini-high | 1063688 | 978891 |
| Gemini-2.5-Flash | 2742487 | 332569 |
| Gemini-2.5-Pro | 2880737 | 455739 |
| LLaMA-3.3 | 969887 | 77523 |
| LLaMA-4-Maverick | 1497730 | 77080 |
| Qwen3-235B | 1089888 | 127447 |

## F  DETAILED ANALYSIS OF CASE STUDY

Here we include a detailed analysis of the case study in Section 5.3. We append the entire code of each iteration and analyze the generated solution.

In the first iteration, the GPT-o4-mini-high model generates an algorithm based on cut enumeration and dynamic programming (DP)-based cut selection, which is the common approach used in technology mapping tools. The core logic is highlighted in the following code snippet. However, it does not prune the cut enumeration space and computes all possible $K$-feasible cuts for each node, which leads to an exponential explosion in computation time. While this achieves high quality, it frequently triggers `TimeoutError` for larger datasets, resulting in low yield.

```python
# iteration 1: base algorithm without pruning the search space

def solve(input_file: str, solution_file: str):
    from collections import deque
    class Node:
        __slots__ = ('name','fanin_names','fanins','fanouts','patterns_one','patterns_zero'
        ↪ ,'const_val')
        def __init__(self,name):
            self.name = name
            self.fanin_names = []
            self.fanins = []
            self.fanouts = []
            self.patterns_one = []
            self.patterns_zero = []
            self.const_val = None
    raw_lines = open(input_file).read().splitlines()
    lines = []
    i = 0
    while i < len(raw_lines):
        line = raw_lines[i].rstrip()
        if not line or line.lstrip().startswith('#'):
            i += 1
            continue
        if line.endswith('\\'):
            merged = line[:-1].rstrip()
            i += 1
            while i < len(raw_lines):
                nxt = raw_lines[i].rstrip()
```

```
28                    if nxt.endswith('\\'):
29                        merged += ' ' + nxt[:-1].rstrip()
30                        i += 1
31                    else:
32                        merged += ' ' + nxt
33                        i += 1
34                        break
35                lines.append(merged)
36            else:
37                lines.append(line)
38                i += 1
39        model_name = ''
40        input_list = []
41        output_list = []
42        nodes = {}
43        idx = 0
44        while idx < len(lines):
45            l = lines[idx].strip()
46            if not l or l.startswith('#'):
47                idx += 1
48                continue
49            if l.startswith('.model'):
50                parts = l.split()
51                if len(parts) > 1:
52                    model_name = parts[1]
53                idx += 1
54            elif l.startswith('.inputs'):
55                parts = l.split()
56                for name in parts[1:]:
57                    input_list.append(name)
58                    if name not in nodes:
59                        nodes[name] = Node(name)
60                idx += 1
61            elif l.startswith('.outputs'):
62                parts = l.split()
63                for name in parts[1:]:
64                    output_list.append(name)
65                idx += 1
66            elif l.startswith('.names'):
67                parts = l.split()
68                in_names = parts[1:-1]
69                out_name = parts[-1]
70                if out_name not in nodes:
71                    nodes[out_name] = Node(out_name)
72                node = nodes[out_name]
73                node.fanin_names = in_names[:]
74                patterns = []
75                j = idx + 1
76                while j < len(lines):
77                    lj = lines[j].strip()
78                    if not lj or lj.startswith('.') or lj.startswith('#'):
79                        break
80                    patterns.append(lj)
81                    j += 1
82                if len(in_names) == 0:
83                    if patterns and patterns[0].strip() == '1':
84                        node.const_val = 1
85                    else:
86                        node.const_val = 0
87                else:
88                    for pline in patterns:
89                        parts2 = pline.split()
90                        if len(parts2) == 1:
91                            mask = parts2[0]; bit = '1'
92                        else:
93                            mask = parts2[0]; bit = parts2[1]
94                        if bit == '1':
95                            node.patterns_one.append(mask)
96                        else:
97                            node.patterns_zero.append(mask)
98                idx = j
99            elif l.startswith('.end'):
100                break
101            else:
102                idx += 1
103        for node in nodes.values():
104            node.fanins = []
105            for fn in node.fanin_names:
106                if fn in nodes:
107                    node.fanins.append(nodes[fn])
108            node.fanouts = []
109        for node in nodes.values():
110            for f in node.fanins:
```

```
111                    f.fanouts.append(node)
112        reachable = set()
113        dq = deque()
114        for out in output_list:
115            if out in nodes:
116                dq.append(out)
117        while dq:
118            u = dq.pop()
119            if u in reachable:
120                continue
121            reachable.add(u)
122            for f in nodes[u].fanins:
123                if f.name not in reachable:
124                    dq.append(f.name)
125        nodes = {name:node for name,node in nodes.items() if name in reachable}
126        for node in nodes.values():
127            node.fanins = [f for f in node.fanins if f.name in nodes]
128            node.fanouts = [f for f in node.fanouts if f.name in nodes]
129        indeg = {name: len(node.fanins) for name,node in nodes.items()}
130        dq = deque([name for name,d in indeg.items() if d == 0])
131        topo_names = []
132        while dq:
133            u = dq.popleft()
134            topo_names.append(u)
135            for w in nodes[u].fanouts:
136                indeg[w.name] -= 1
137                if indeg[w.name] == 0:
138                    dq.append(w.name)
139        topo_list = [nodes[name] for name in topo_names]
140        K = 6
141        def prune_cuts(cset):
142            cuts = list(cset)
143            res = []
144            for c in cuts:
145                skip = False
146                for d in cuts:
147                    if d is not c and d.issubset(c):
148                        skip = True
149                        break
150                if not skip:
151                    res.append(c)
152            return res
153        cuts = {}        # Cut Enumeration
154        for n in topo_list:
155            if not n.fanins:
156                cuts[n.name] = [frozenset([n.name])]
157            else:
158                cuts_n = None
159                for f in n.fanins:
160                    cf = cuts[f.name]
161                    if cuts_n is None:
162                        cuts_n = cf[:]
163                    else:
164                        newset = set()
165                        for c1 in cuts_n:
166                            for c2 in cf:
167                                u = c1 | c2
168                                if len(u) <= K:
169                                    newset.add(u)
170                        cuts_n = prune_cuts(newset)
171                if cuts_n is None:
172                    cuts_n = []
173                cuts_set = set(cuts_n)
174                cuts_set.add(frozenset([n.name]))
175                cuts[n.name] = prune_cuts(cuts_set)
176        cost = {}
177        best_cut = {}
178        INF = 10**18
179        for n in topo_list:    # DP-based Cut Selection
180            if not n.fanins:
181                cost[n.name] = 0
182                best_cut[n.name] = None
183            else:
184                bcut = None
185                bval = INF
186                for c in cuts[n.name]:
187                    if c == frozenset([n.name]):
188                        continue
189                    s = 0
190                    for m in c:
191                        s += cost.get(m,0)
192                    cval = s + 1
193                    if cval < bval:
```

```
194                         bval = cval; bcut = c
195                 if bcut is None:
196                     fl = frozenset([f.name for f in n.fanins])
197                     bcut = fl
198                     s = 0
199                     for m in bcut:
200                         s += cost.get(m,0)
201                     bval = s + 1
202                 cost[n.name] = bval
203                 best_cut[n.name] = bcut
204         mapping_nodes = set()
205         dq = deque()
206         for out in output_list:
207             if out in best_cut and best_cut[out] is not None:
208                 dq.append(out)
209         while dq:
210             u = dq.pop()
211             if u in mapping_nodes:
212                 continue
213             mapping_nodes.add(u)
214             c = best_cut[u]
215             if c:
216                 for m in c:
217                     if m in nodes:
218                         nm = nodes[m]
219                         if m not in input_list and nm.const_val is None:
220                             dq.append(m)
221         mapping_topo = [n for n in topo_list if n.name in mapping_nodes]
222         f = open(solution_file,'w')
223         f.write('.model '+model_name+'\n')
224         f.write('.inputs '+' '.join(input_list)+'\n')
225         f.write('.outputs '+' '.join(output_list)+'\n')
226         for n in mapping_topo:
227             leaves = best_cut[n.name]
228             inputs = sorted(leaves)
229             k = len(inputs)
230             N = 1<<k
231             full_mask = (1<<N) - 1
232             masks = []
233             for i in range(k):
234                 m = 0
235                 for j in range(N):
236                     if (j>>i)&1:
237                         m |= (1<<j)
238                 masks.append(m)
239             vis = set()
240             queue2 = [n.name]
241             while queue2:
242                 u = queue2.pop()
243                 if u in vis or u in inputs:
244                     continue
245                 vis.add(u)
246                 for fn in nodes[u].fanins:
247                     if fn.name not in vis:
248                         queue2.append(fn.name)
249             cone_nodes = [nodes[name] for name in topo_names if name in vis]
250             val = {}
251             for i,name in enumerate(inputs):
252                 val[name] = masks[i]
253             for v in cone_nodes:
254                 if v.const_val is not None:
255                     val[v.name] = full_mask if v.const_val==1 else 0
256                 else:
257                     if v.patterns_zero:
258                         zero_mask = 0
259                         for p in v.patterns_zero:
260                             m = full_mask
261                             for idx,ch in enumerate(p):
262                                 vm = val[v.fanin_names[idx]]
263                                 if ch=='1':
264                                     m &= vm
265                                 elif ch=='0':
266                                     m &= (~vm)&full_mask
267                             zero_mask |= m
268                         val[v.name] = (~zero_mask)&full_mask
269                     else:
270                         one_mask = 0
271                         for p in v.patterns_one:
272                             m = full_mask
273                             for idx,ch in enumerate(p):
274                                 vm = val[v.fanin_names[idx]]
275                                 if ch=='1':
276                                     m &= vm
```

```
277                        elif ch=='0':
278                            m &= (~vm)&full_mask
279                    one_mask |= m
280                val[v.name] = one_mask
281        root_mask = val[n.name]
282        f.write('.names '+' '.join(inputs)+' '+n.name+'\n')
283        for j in range(N):
284            if (root_mask>>j)&1:
285                line = ''.join('1' if (j>>i)&1 else '0' for i in range(k))
286                f.write(line+' 1\n')
287    f.write('.end\n')
288    f.close()
289
```

In the second iteration, the model overcorrects by adopting a naive 6-LUT mapping strategy that trivially assigns each node to a separate LUT. While this approach passes all test cases within the timeout, it produces poor solution quality. Specifically, it redundantly maps each non-primary-input node as an individual LUT by directly copying inputs to outputs, resulting in excessive and inefficient resource usage.

```
1   # iteration 2: a very naive mapping strategy. Simply copy the input to the output.
2
3   def solve(input_file: str, solution_file: str):
4       raw = open(input_file).read().splitlines()
5       merged = []
6       i = 0
7       while i < len(raw):
8           line = raw[i].rstrip()
9           if not line or line.lstrip().startswith('#'):
10              i += 1
11              continue
12          if line.endswith('\\'):
13              accum = line[:-1].rstrip()
14              i += 1
15              while i < len(raw):
16                  nxt = raw[i].rstrip()
17                  if nxt.endswith('\\'):
18                      accum += ' ' + nxt[:-1].rstrip()
19                      i += 1
20                  else:
21                      accum += ' ' + nxt
22                      i += 1
23                      break
24              merged.append(accum)
25          else:
26              merged.append(line)
27              i += 1
28      with open(solution_file, 'w') as f:
29          for l in merged:
30              f.write(l + '\n')
```

In the third iteration, the model proposes a more sophisticated strategy that first applies static pruning to the cut enumeration space – limiting the number of cuts per node to 20 – followed by a dynamic programming-based cut selection. This approach improves solution quality, though full yield is still not achieved. Additionally, truth table computation is optimized using bitwise operations.

```
1   # iteration 3: static pruning by capping the number of cuts per node to M = 20; efficient
    ↪  bitwise operations
2
3   def solve(input_file: str, solution_file: str):
4       from collections import deque
5       raw_lines = open(input_file).read().splitlines()
6       lines = []
7       i = 0
8       while i < len(raw_lines):
9           line = raw_lines[i].rstrip()
10          if not line or line.lstrip().startswith('#'):
11              i += 1
12              continue
13          if line.endswith('\\'):
14              accum = line[:-1].rstrip()
15              i += 1
16              while i < len(raw_lines):
17                  nxt = raw_lines[i].rstrip()
18                  if nxt.endswith('\\'):
19                      accum += ' ' + nxt[:-1].rstrip()
```

```
20                        i += 1
21                    else:
22                        accum += ' ' + nxt
23                        i += 1
24                        break
25                lines.append(accum)
26            else:
27                lines.append(line)
28                i += 1
29    class Node:
30        def __init__(self, name):
31            self.name = name
32            self.fanin_names = []
33            self.patterns_one = []
34            self.patterns_zero = []
35            self.const_val = None
36            self.fanins = []
37            self.fanouts = []
38    model_name = ''
39    input_list = []
40    output_list = []
41    nodes = {}
42    idx = 0
43    while idx < len(lines):
44        l = lines[idx].strip()
45        if not l:
46            idx += 1
47            continue
48        if l.startswith('.model'):
49            parts = l.split()
50            if len(parts) > 1:
51                model_name = parts[1]
52            idx += 1
53        elif l.startswith('.inputs'):
54            parts = l.split()
55            for name in parts[1:]:
56                input_list.append(name)
57                if name not in nodes:
58                    nodes[name] = Node(name)
59            idx += 1
60        elif l.startswith('.outputs'):
61            parts = l.split()
62            for name in parts[1:]:
63                output_list.append(name)
64            idx += 1
65        elif l.startswith('.names'):
66            parts = l.split()
67            in_names = parts[1:-1]
68            out_name = parts[-1]
69            if out_name not in nodes:
70                nodes[out_name] = Node(out_name)
71            node = nodes[out_name]
72            node.fanin_names = in_names[:]
73            patterns = []
74            j = idx + 1
75            while j < len(lines) and not lines[j].startswith('.'):
76                s = lines[j].strip()
77                if s:
78                    patterns.append(s)
79                j += 1
80            if len(in_names) == 0:
81                if patterns and patterns[0] == '1':
82                    node.const_val = 1
83                else:
84                    node.const_val = 0
85            else:
86                for pline in patterns:
87                    parts2 = pline.split()
88                    if len(parts2) == 1:
89                        mask = parts2[0]; bit = '1'
90                    else:
91                        mask = parts2[0]; bit = parts2[1]
92                    if bit == '1':
93                        node.patterns_one.append(mask)
94                    else:
95                        node.patterns_zero.append(mask)
96            idx = j
97        elif l.startswith('.end'):
98            break
99        else:
100            idx += 1
101    for node in nodes.values():
102        node.fanins = [nodes[nm] for nm in node.fanin_names if nm in nodes]
```

```
103          node.fanouts = []
104      for node in nodes.values():
105          for f in node.fanins:
106              f.fanouts.append(node)
107      reachable = set()
108      stack = [nm for nm in output_list if nm in nodes]
109      while stack:
110          u = stack.pop()
111          if u in reachable: continue
112          reachable.add(u)
113          for p in nodes[u].fanin_names:
114              if p in nodes and p not in reachable:
115                  stack.append(p)
116      nodes = {nm:nd for nm,nd in nodes.items() if nm in reachable}
117      for nd in nodes.values():
118          nd.fanins = [f for f in nd.fanins if f.name in nodes]
119          nd.fanouts = [f for f in nd.fanouts if f.name in nodes]
120      indeg = {nm: len(nd.fanins) for nm,nd in nodes.items()}
121      dq = deque([nm for nm,d in indeg.items() if d == 0])
122      topo = []
123      while dq:
124          u = dq.popleft()
125          topo.append(u)
126          for w in nodes[u].fanouts:
127              indeg[w.name] -= 1
128              if indeg[w.name] == 0:
129                  dq.append(w.name)
130      topo_nodes = [nodes[nm] for nm in topo]
131      name_to_idx = {nd.name:i for i,nd in enumerate(topo_nodes)}
132      idx_to_node = topo_nodes
133      idx_to_name = [nd.name for nd in topo_nodes]
134      N = len(topo_nodes)
135      PI_idx = set(name_to_idx[nm] for nm in input_list if nm in name_to_idx)
136      self_mask = [1 << i for i in range(N)]
137      K = 6
138      M = 20
139      cuts = [[] for _ in range(N)]
140      for i, nd in enumerate(topo_nodes):
141          if i in PI_idx or nd.const_val is not None:
142              cuts[i] = [self_mask[i]]
143              continue
144          fan_idxs = [name_to_idx[x] for x in nd.fanin_names if x in name_to_idx]
145          c_list = None
146          for f in fan_idxs:
147              fcuts = cuts[f]
148              if c_list is None:
149                  c_list = fcuts[:M]
150              else:
151                  newset = set()
152                  for a in c_list[:M]:
153                      for b in fcuts[:M]:
154                          u = a | b
155                          if u.bit_count() <= K:
156                              newset.add(u)
157                  if newset:
158                      lst = sorted(newset, key=lambda x: x.bit_count())
159                      c_list = lst[:M]
160                  else:
161                      c_list = []
162              if not c_list:
163                  break
164          if c_list is None:
165              c_list = []
166          s = set(c_list)
167          s.add(self_mask[i])
168          lst2 = sorted(s, key=lambda x: x.bit_count())
169          cuts[i] = lst2[:M]
170      INF = 10**18
171      cost = [0]*N
172      bestM = [None]*N
173      for i, nd in enumerate(topo_nodes):
174          if i in PI_idx or nd.const_val is not None:
175              cost[i] = 0
176              bestM[i] = None
177              continue
178          bestc = INF
179          bm = None
180          for cm in cuts[i]:
181              if cm == self_mask[i]:
182                  continue
183              s = 0
184              m = cm
185              while m:
```

```
186                    lsb = m & -m
187                    j = lsb.bit_length() - 1
188                    s += cost[j]
189                    m ^= lsb
190                cval = s + 1
191                if cval < bestc:
192                    bestc = cval
193                    bm = cm
194            if bm is None:
195                s = 0
196                mask = 0
197                for f in nd.fanin_names:
198                    if f in name_to_idx:
199                        j = name_to_idx[f]
200                        mask |= self_mask[j]
201                        s += cost[j]
202                bm = mask
203                bestc = s + 1
204            cost[i] = bestc
205            bestM[i] = bm
206        mapping = set()
207        stack = [name_to_idx[nm] for nm in output_list if nm in name_to_idx]
208        visited = set()
209        while stack:
210            u = stack.pop()
211            if u in visited:
212                continue
213            visited.add(u)
214            bm = bestM[u]
215            if bm is None or bm == self_mask[u]:
216                continue
217            mapping.add(u)
218            m = bm
219            while m:
220                lsb = m & -m
221                j = lsb.bit_length() - 1
222                m ^= lsb
223                if j not in visited and j not in PI_idx and topo_nodes[j].const_val is None:
224                    stack.append(j)
225        mapped = [i for i in range(N) if i in mapping]
226        with open(solution_file, 'w') as f:
227            f.write('.model ' + model_name + '\n')
228            f.write('.inputs ' + ' '.join(input_list) + '\n')
229            f.write('.outputs ' + ' '.join(output_list) + '\n')
230            for i in mapped:
231                nd = topo_nodes[i]
232                bm = bestM[i]
233                leaves = [j for j in range(N) if (bm >> j) & 1]
234                inputs = [idx_to_name[j] for j in leaves]
235                k = len(inputs)
236                Np = 1 << k
237                full = (1 << Np) - 1
238                masks = []
239                for t in range(k):
240                    mm = 0
241                    for j in range(Np):
242                        if (j >> t) & 1:
243                            mm |= 1 << j
244                    masks.append(mm)
245                val = {}
246                for t,j in enumerate(leaves):
247                    val[j] = masks[t]
248                vis = set(leaves)
249                stack2 = [i]
250                while stack2:
251                    u = stack2.pop()
252                    if u in vis:
253                        continue
254                    vis.add(u)
255                    for p in topo_nodes[u].fanins:
256                        pi = name_to_idx.get(p.name)
257                        if pi is not None and pi not in vis:
258                            stack2.append(pi)
259                cone = [j for j in range(N) if j in vis and j not in leaves]
260                for j in sorted(cone, key=lambda x: name_to_idx[topo_nodes[x].name]):
261                    v = topo_nodes[j]
262                    if v.const_val is not None:
263                        val[j] = full if v.const_val == 1 else 0
264                    else:
265                        if v.patterns_zero:
266                            zm = 0
267                            for p in v.patterns_zero:
268                                mm = full
```

```
269                              for t,ch in enumerate(p):
270                                  fn = v.fanin_names[t]
271                                  pj = name_to_idx.get(fn)
272                                  if pj is None:
273                                      mm = 0
274                                      break
275                                  vm = val[pj]
276                                  if ch == '1':
277                                      mm &= vm
278                                  elif ch == '0':
279                                      mm &= (~vm) & full
280                              zm |= mm
281                          val[j] = (~zm) & full
282                      else:
283                          om = 0
284                          for p in v.patterns_one:
285                              mm = full
286                              for t,ch in enumerate(p):
287                                  fn = v.fanin_names[t]
288                                  pj = name_to_idx.get(fn)
289                                  if pj is None:
290                                      mm = 0
291                                      break
292                                  vm = val[pj]
293                                  if ch == '1':
294                                      mm &= vm
295                                  elif ch == '0':
296                                      mm &= (~vm) & full
297                              om |= mm
298                          val[j] = om
299                  root = val[i]
300                  f.write('.names ' + ' '.join(inputs) + ' ' + nd.name + '\n')
301                  for j in range(Np):
302                      if (root >> j) & 1:
303                          bits = ''.join('1' if (j >> t) & 1 else '0' for t in range(k))
304                          f.write(bits + ' 1\n')
305              f.write('.end\n')
306
```

In the fourth iteration, the cut limit per node is increased from 20 to 30, enabling broader solution exploration and potentially improving quality. Additionally, the algorithm reduces redundant computations by caching precomputed scores for each cut.

```
1   # iteration 4: explore larger solution space; reduce redundant computations
2
3   def solve(input_file: str, solution_file: str):
4       from collections import deque
5       raw = open(input_file).read().splitlines()
6       lines = []
7       i = 0
8       while i < len(raw):
9           l = raw[i].rstrip()
10          if not l or l.lstrip().startswith('#'):
11              i += 1
12              continue
13          if l.endswith('\\'):
14              acc = l[:-1].rstrip()
15              i += 1
16              while i < len(raw):
17                  nl = raw[i].rstrip()
18                  if nl.endswith('\\'):
19                      acc += ' ' + nl[:-1].rstrip()
20                      i += 1
21                  else:
22                      acc += ' ' + nl
23                      i += 1
24                      break
25              lines.append(acc)
26          else:
27              lines.append(l)
28              i += 1
29      class Node:
30          __slots__ = ('name','fanin_names','patterns_one','patterns_zero','const_val','fanin
         ↪ s','fanouts')
31          def __init__(self,n):
32              self.name = n
33              self.fanin_names = []
34              self.patterns_one = []
35              self.patterns_zero = []
36              self.const_val = None
```

```
37                    self.fanins = []
38                    self.fanouts = []
39        model = ''
40        inputs = []
41        outputs = []
42        nodes = {}
43        idx = 0
44        while idx < len(lines):
45            l = lines[idx].strip()
46            if not l:
47                idx += 1; continue
48            if l.startswith('.model'):
49                parts = l.split()
50                if len(parts)>1: model = parts[1]
51                idx += 1
52            elif l.startswith('.inputs'):
53                parts = l.split()
54                for nm in parts[1:]:
55                    inputs.append(nm)
56                    if nm not in nodes: nodes[nm] = Node(nm)
57                idx += 1
58            elif l.startswith('.outputs'):
59                parts = l.split()
60                for nm in parts[1:]:
61                    outputs.append(nm)
62                idx += 1
63            elif l.startswith('.names'):
64                parts = l.split()
65                inps = parts[1:-1]; outp = parts[-1]
66                if outp not in nodes: nodes[outp] = Node(outp)
67                nd = nodes[outp]
68                nd.fanin_names = inps[:]
69                pats = []
70                j = idx+1
71                while j < len(lines) and not lines[j].startswith('.'):
72                    s = lines[j].strip()
73                    if s: pats.append(s)
74                    j += 1
75                if not inps:
76                    if pats and pats[0]=='1': nd.const_val = 1
77                    else: nd.const_val = 0
78                else:
79                    for pt in pats:
80                        sp = pt.split()
81                        if len(sp)==1:
82                            mask = sp[0]; bit = '1'
83                        else:
84                            mask,bit = sp[0],sp[1]
85                        if bit=='1': nd.patterns_one.append(mask)
86                        else: nd.patterns_zero.append(mask)
87                idx = j
88            elif l.startswith('.end'):
89                break
90            else:
91                idx += 1
92        for nd in nodes.values():
93            nd.fanins = [nodes[nm] for nm in nd.fanin_names if nm in nodes]
94        for nd in nodes.values():
95            for f in nd.fanins:
96                f.fanouts.append(nd)
97        reachable = set()
98        st = [nm for nm in outputs if nm in nodes]
99        while st:
100            u = st.pop()
101            if u in reachable: continue
102            reachable.add(u)
103            for p in nodes[u].fanin_names:
104                if p in nodes and p not in reachable:
105                    st.append(p)
106        nodes = {nm:nd for nm,nd in nodes.items() if nm in reachable}
107        for nd in nodes.values():
108            nd.fanins = [f for f in nd.fanins if f.name in nodes]
109            nd.fanouts = [f for f in nd.fanouts if f.name in nodes]
110        indeg = {nm: len(nd.fanins) for nm,nd in nodes.items()}
111        dq = deque([nm for nm,d in indeg.items() if d==0])
112        topo = []
113        while dq:
114            u = dq.popleft(); topo.append(u)
115            for w in nodes[u].fanouts:
116                indeg[w.name] -= 1
117                if indeg[w.name]==0: dq.append(w.name)
118        topo_nodes = [nodes[nm] for nm in topo]
119        N = len(topo_nodes)
```

```
120        name_to_idx = {nd.name:i for i,nd in enumerate(topo_nodes)}
121        idx_to_name = [nd.name for nd in topo_nodes]
122        PI = set(name_to_idx[nm] for nm in inputs if nm in name_to_idx)
123        cost = [0]*N
124        bestM = [None]*N
125        K = 6
126        M = 30
127        cuts = [[] for _ in range(N)]
128        for i,nd in enumerate(topo_nodes):
129            if i in PI or nd.const_val is not None:
130                cost[i] = 0
131                bestM[i] = None
132                cuts[i] = [(1<<i, 0)]
133                continue
134            fans = [name_to_idx[nm] for nm in nd.fanin_names if nm in name_to_idx]
135            c_list = None
136            for f in fans:
137                fcuts = cuts[f]
138                if len(fcuts) > M: fcuts = fcuts[:M]
139                if c_list is None:
140                    c_list = fcuts.copy()
141                else:
142                    newm = {}
143                    for m1,sc1 in c_list:
144                        for m2,sc2 in fcuts:
145                            m = m1 | m2
146                            if m.bit_count() <= K:
147                                if m in newm: continue
148                                t = m; sc = 0
149                                while t:
150                                    lsb = t & -t; j = lsb.bit_length()-1
151                                    sc += cost[j]; t ^= lsb
152                                newm[m] = sc
153                    if not newm:
154                        c_list = []
155                        break
156                    items = sorted(newm.items(), key=lambda x: x[1])
157                    c_list = items[:M]
158            if c_list is None: c_list = []
159            fb = 0
160            scf = 0
161            for f in fans:
162                fb |= (1<<f)
163                scf += cost[f]
164            if fb.bit_count() <= K:
165                if not any(m==fb for m,_ in c_list):
166                    c_list.append((fb, scf))
167            bestc = 10**18; bm = None
168            for m,sc in c_list:
169                v = sc + 1
170                if v < bestc:
171                    bestc = v; bm = m
172            if bm is None:
173                bm = fb; bestc = scf + 1
174            cost[i] = bestc; bestM[i] = bm
175            cuts[i] = sorted(c_list, key=lambda x: x[1])[:M]
176        mapping = set()
177        st = [name_to_idx[nm] for nm in outputs if nm in name_to_idx]
178        vis = set()
179        while st:
180            u = st.pop()
181            if u in vis: continue
182            vis.add(u)
183            bm = bestM[u]
184            if bm is None: continue
185            mapping.add(u)
186            t = bm
187            while t:
188                lsb = t & -t; j = lsb.bit_length()-1; t ^= lsb
189                if j in PI or topo_nodes[j].const_val is not None: continue
190                if j not in vis: st.append(j)
191        mapped = sorted(mapping)
192        with open(solution_file, 'w') as f:
193            f.write('.model ' + model + '\n')
194            f.write('.inputs ' + ' '.join(inputs) + '\n')
195            f.write('.outputs ' + ' '.join(outputs) + '\n')
196            for i in mapped:
197                nd = topo_nodes[i]
198                bm = bestM[i]
199                leaves = [j for j in range(N) if (bm>>j)&1]
200                inputs_l = [idx_to_name[j] for j in leaves]
201                k = len(inputs_l)
202                Np = 1<<k
```

```
203                    full = (1<<Np) - 1
204                    masks = []
205                    for t in range(k):
206                        mm = 0
207                        for j in range(Np):
208                            if (j>>t)&1: mm |= 1<<j
209                        masks.append(mm)
210                    val = {}
211                    for t,j in enumerate(leaves):
212                        val[j] = masks[t]
213                    vis2 = set(leaves)
214                    stk = [i]
215                    while stk:
216                        u = stk.pop()
217                        if u in vis2: continue
218                        vis2.add(u)
219                        for p in topo_nodes[u].fanins:
220                            pj = name_to_idx.get(p.name)
221                            if pj is not None and pj not in vis2:
222                                stk.append(pj)
223                    cone = [j for j in topo if False]
224                    cone = [j for j in range(N) if j in vis2 and j not in leaves]
225                    for j in sorted(cone):
226                        v = topo_nodes[j]
227                        if v.const_val is not None:
228                            val[j] = full if v.const_val==1 else 0
229                        else:
230                            if v.patterns_zero:
231                                zm = 0
232                                for p in v.patterns_zero:
233                                    mm = full
234                                    for t,ch in enumerate(p):
235                                        fn = v.fanin_names[t]
236                                        pj = name_to_idx.get(fn)
237                                        if pj is None:
238                                            mm = 0; break
239                                        vm = val[pj]
240                                        if ch=='1': mm &= vm
241                                        elif ch=='0': mm &= (~vm)&full
242                                    zm |= mm
243                                val[j] = (~zm)&full
244                            else:
245                                om = 0
246                                for p in v.patterns_one:
247                                    mm = full
248                                    for t,ch in enumerate(p):
249                                        fn = v.fanin_names[t]
250                                        pj = name_to_idx.get(fn)
251                                        if pj is None:
252                                            mm = 0; break
253                                        vm = val[pj]
254                                        if ch=='1': mm &= vm
255                                        elif ch=='0': mm &= (~vm)&full
256                                    om |= mm
257                                val[j] = om
258                    root = val[i]
259                    f.write('.names ' + ' '.join(inputs_l) + ' ' + nd.name + '\n')
260                    for j in range(Np):
261                        if (root>>j)&1:
262                            pat = ''.join('1' if (j>>t)&1 else '0' for t in range(k))
263                            f.write(pat + ' 1\n')
264              f.write('.end\n')
265
```

In the fifth iteration, it introduces a dynamic pruning method by adapting the cut limit based on circuit size—more cuts for smaller circuits (higher quality), and fewer for large circuits (better yield).

```
1   # iteration 5: dynamic pruning of cut enumeration space
2
3   def solve(input_file: str, solution_file: str):
4       from collections import deque
5       raw = open(input_file).read().splitlines()
6       lines = []
7       i = 0
8       while i < len(raw):
9           l = raw[i].rstrip()
10          if not l or l.lstrip().startswith('#'):
11              i += 1
12              continue
13          if l.endswith('\\'):
14              acc = l[:-1].rstrip()
```

```python
15                    i += 1
16                while i < len(raw):
17                    nl = raw[i].rstrip()
18                    if nl.endswith('\\'):
19                        acc += ' ' + nl[:-1].rstrip()
20                        i += 1
21                    else:
22                        acc += ' ' + nl
23                        i += 1
24                        break
25                lines.append(acc)
26            else:
27                lines.append(l)
28            i += 1
29    class Node:
30        __slots__ = ('name','fanin_names','patterns_one','patterns_zero','const_val','fanin
        ↪ s','fanouts')
31        def __init__(self,n):
32            self.name = n
33            self.fanin_names = []
34            self.patterns_one = []
35            self.patterns_zero = []
36            self.const_val = None
37            self.fanins = []
38            self.fanouts = []
39    model = ''
40    inputs = []
41    outputs = []
42    nodes = {}
43    idx = 0
44    while idx < len(lines):
45        l = lines[idx].strip()
46        if not l:
47            idx += 1; continue
48        if l.startswith('.model'):
49            parts = l.split()
50            if len(parts) > 1: model = parts[1]
51            idx += 1
52        elif l.startswith('.inputs'):
53            parts = l.split()
54            for nm in parts[1:]:
55                inputs.append(nm)
56                if nm not in nodes: nodes[nm] = Node(nm)
57            idx += 1
58        elif l.startswith('.outputs'):
59            parts = l.split()
60            for nm in parts[1:]:
61                outputs.append(nm)
62            idx += 1
63        elif l.startswith('.names'):
64            parts = l.split()
65            inps = parts[1:-1]; outp = parts[-1]
66            if outp not in nodes: nodes[outp] = Node(outp)
67            nd = nodes[outp]
68            nd.fanin_names = inps[:]
69            pats = []
70            j = idx + 1
71            while j < len(lines) and not lines[j].startswith('.'):
72                s = lines[j].strip()
73                if s: pats.append(s)
74                j += 1
75            if not inps:
76                nd.const_val = 1 if pats and pats[0] == '1' else 0
77            else:
78                for pt in pats:
79                    sp = pt.split()
80                    if len(sp) == 1:
81                        mask, bit = sp[0], '1'
82                    else:
83                        mask, bit = sp[0], sp[1]
84                    if bit == '1': nd.patterns_one.append(mask)
85                    else: nd.patterns_zero.append(mask)
86            idx = j
87        elif l.startswith('.end'):
88            break
89        else:
90            idx += 1
91    for nd in nodes.values():
92        nd.fanins = [nodes[nm] for nm in nd.fanin_names if nm in nodes]
93    for nd in nodes.values():
94        for f in nd.fanins:
95            f.fanouts.append(nd)
96    reachable = set()
```

```
97          st = [nm for nm in outputs if nm in nodes]
98          while st:
99              u = st.pop()
100             if u in reachable: continue
101             reachable.add(u)
102             for p in nodes[u].fanin_names:
103                 if p in nodes and p not in reachable:
104                     st.append(p)
105         nodes = {nm:nd for nm,nd in nodes.items() if nm in reachable}
106         for nd in nodes.values():
107             nd.fanins = [f for f in nd.fanins if f.name in nodes]
108             nd.fanouts = [f for f in nd.fanouts if f.name in nodes]
109         indeg = {nm: len(nd.fanins) for nm,nd in nodes.items()}
110         dq = deque([nm for nm,d in indeg.items() if d == 0])
111         topo = []
112         while dq:
113             u = dq.popleft(); topo.append(u)
114             for w in nodes[u].fanouts:
115                 indeg[w.name] -= 1
116                 if indeg[w.name] == 0: dq.append(w.name)
117         topo_nodes = [nodes[nm] for nm in topo]
118         N = len(topo_nodes)
119         name_to_idx = {nd.name: i for i, nd in enumerate(topo_nodes)}
120         idx_to_name = [nd.name for nd in topo_nodes]
121         PI = set(name_to_idx[nm] for nm in inputs if nm in name_to_idx)
122         cost = [0] * N
123         bestM = [None] * N
124         K = 6
125         if N <= 1500:
126             M = 64
127         elif N <= 3000:
128             M = 48
129         elif N <= 5000:
130             M = 32
131         else:
132             M = 20
133         cuts = [[] for _ in range(N)]
134         def prune_cuts(items, limit):
135             items_sorted = sorted(items, key=lambda x: (x[1], x[0].bit_count()))
136             pr = []
137             for m, sc in items_sorted:
138                 dom = False
139                 for pm, psc in pr:
140                     if psc <= sc and (pm & m) == pm:
141                         dom = True
142                         break
143                 if not dom:
144                     pr.append((m, sc))
145                     if len(pr) >= limit:
146                         break
147             return pr
148         for i, nd in enumerate(topo_nodes):
149             if i in PI or nd.const_val is not None:
150                 cost[i] = 0
151                 bestM[i] = None
152                 cuts[i] = [(1 << i, 0)]
153                 continue
154             fans = [name_to_idx[nm] for nm in nd.fanin_names if nm in name_to_idx]
155             fans.sort(key=lambda x: len(cuts[x]))
156             c_list = None
157             for f in fans:
158                 fcuts = cuts[f]
159                 if not fcuts:
160                     c_list = []
161                     break
162                 fcuts = fcuts[:M]
163                 if c_list is None:
164                     c_list = fcuts.copy()
165                 else:
166                     newm = {}
167                     for m1, sc1 in c_list:
168                         for m2, sc2 in fcuts:
169                             m = m1 | m2
170                             if m.bit_count() <= K:
171                                 s2 = sc1 + sc2
172                                 prev = newm.get(m)
173                                 if prev is None or s2 < prev:
174                                     newm[m] = s2
175                     if not newm:
176                         c_list = []
177                         break
178                     c_list = prune_cuts(list(newm.items()), M)
179             if not c_list:
```

```
180                    um = 0; usc = 0
181                    for f in fans:
182                        um |= (1 << f)
183                        usc += cost[f]
184                    c_list = [(um, usc)]
185                um = 0; usc = 0
186                for f in fans:
187                    um |= (1 << f)
188                    usc += cost[f]
189                if um.bit_count() <= K and all(m != um for m, _ in c_list):
190                    c_list.append((um, usc))
191                c_list = prune_cuts(c_list, M)
192                bestc = 10**18; bm = None
193                selfm = (1 << i)
194                for m, sc in c_list:
195                    if m == selfm:
196                        continue
197                    v = sc + 1
198                    if v < bestc:
199                        bestc = v; bm = m
200                if bm is None:
201                    bm = um
202                    bestc = usc + 1
203                cost[i] = bestc
204                bestM[i] = bm
205                cuts[i] = c_list
206        mapping = set()
207        st = [name_to_idx[nm] for nm in outputs if nm in name_to_idx]
208        vis = set()
209        while st:
210            u = st.pop()
211            if u in vis: continue
212            vis.add(u)
213            bm = bestM[u]
214            if bm is None: continue
215            mapping.add(u)
216            t = bm
217            while t:
218                lsb = t & -t
219                j = lsb.bit_length() - 1
220                t ^= lsb
221                if j in PI or topo_nodes[j].const_val is not None: continue
222                if j not in vis:
223                    st.append(j)
224        mapped = sorted(mapping)
225        with open(solution_file, 'w') as f:
226            f.write('.model ' + model + '\n')
227            f.write('.inputs ' + ' '.join(inputs) + '\n')
228            f.write('.outputs ' + ' '.join(outputs) + '\n')
229            for i in mapped:
230                nd = topo_nodes[i]
231                bm = bestM[i]
232                leaves = [j for j in range(N) if (bm >> j) & 1]
233                inputs_l = [idx_to_name[j] for j in leaves]
234                k = len(inputs_l)
235                Np = 1 << k
236                full = (1 << Np) - 1
237                masks = []
238                for t in range(k):
239                    mm = 0
240                    for j in range(Np):
241                        if (j >> t) & 1:
242                            mm |= 1 << j
243                    masks.append(mm)
244                val = {}
245                for t, j in enumerate(leaves):
246                    val[j] = masks[t]
247                seen = set(leaves)
248                stk = [i]
249                cone = []
250                while stk:
251                    u = stk.pop()
252                    if u in seen: continue
253                    seen.add(u)
254                    cone.append(u)
255                    for p in topo_nodes[u].fanins:
256                        pj = name_to_idx.get(p.name)
257                        if pj is not None and pj not in seen:
258                            stk.append(pj)
259                cone.sort()
260                for j in cone:
261                    v = topo_nodes[j]
262                    if v.const_val is not None:
```

```
263                        val[j] = full if v.const_val == 1 else 0
264                else:
265                    if v.patterns_zero:
266                        zm = 0
267                        for p in v.patterns_zero:
268                            mm = full
269                            for t, ch in enumerate(p):
270                                fn = v.fanin_names[t]
271                                pj = name_to_idx.get(fn)
272                                if pj is None:
273                                    mm = 0
274                                    break
275                                vm = val[pj]
276                                if ch == '1':
277                                    mm &= vm
278                                elif ch == '0':
279                                    mm &= (~vm) & full
280                            zm |= mm
281                        val[j] = (~zm) & full
282                    else:
283                        om = 0
284                        for p in v.patterns_one:
285                            mm = full
286                            for t, ch in enumerate(p):
287                                fn = v.fanin_names[t]
288                                pj = name_to_idx.get(fn)
289                                if pj is None:
290                                    mm = 0
291                                    break
292                                vm = val[pj]
293                                if ch == '1':
294                                    mm &= vm
295                                elif ch == '0':
296                                    mm &= (~vm) & full
297                            om |= mm
298                        val[j] = om
299            root = val[i]
300            f.write('.names ' + ' '.join(inputs_l) + ' ' + nd.name + '\n')
301            for j in range(Np):
302                if (root >> j) & 1:
303                    pat = ''.join('1' if (j >> t) & 1 else '0' for t in range(k))
304                    f.write(pat + ' 1\n')
305        f.write('.end\n')
306
```

## G  DATASETS

We summarize the original data sources for each problem in Table 27 and the number of data instances in Table 28. All datasets are derived from real-world applications. We further partition or transform them into standardized input formats, ensuring the inclusion of both small-scale instances for demonstration purposes and large-scale instances for evaluation. For detailed data organization, please refer to our repository.

Table 27: Datasets used in our benchmark.

| Problem | Original Data Source |
|---------|---------------------|
| Operator scheduling | EXPRESS (Wang et al., 2007) |
| Technology mapping | EPFL (Amarú et al., 2015) and ISCAS85 (Hansen et al., 1999) |
| Global routing | ISPD'24 Contest (Liang et al., 2024) |
| E-graph extraction | SmoothE (Cai et al., 2025) |
| Intra-op parallelism | ASPLOS'24 Contest (Moffitt & Fegade, 2025) |
| Protein sequence design | Protein Data Bank (PDB) (Database, 2024) |
| Mendelian error detection | Cost Function Library (Sanchez et al., 2008; Schiex, 2018) |
| Airline crew pairing | China Graduate Mathematical Modeling Competition'21 F (Competition, 2021) |
| Pickup and delivery w/ time windows | MetaPDPTW (Li & Lim, 2001) |

Table 28: Number of instances of each problem in HeuriGym.

| Problem | # of Instances |
|---|---|
| Operator scheduling | 24 |
| Technology mapping | 31 |
| Global routing | 24 |
| E-graph extraction | 23 |
| Intra-op parallelism | 28 |
| Protein sequence design | 24 |
| Mendelian error detection | 20 |
| Airline crew pairing | 14 |
| Pickup and delivery w/ time windows | 30 |
| **Total** | 218 |

