# OpenReview forum: "HeuriGym: An Agentic Benchmark for LLM-Crafted Heuristics in Combinatorial Optimization"
_ICLR.cc/2026/Conference — ICLR 2026 Poster_

### Official Review · Reviewer_QPGo · 2025-10-19

**Soundness:** 3
**Presentation:** 3
**Contribution:** 2
**Rating:** 4
**Confidence:** 5

**Summary:**

The paper introduces HeuriGym, an agentic framework designed to systematically evaluate large language models (LLMs) on heuristic generation tasks within combinatorial optimization. Its major contributions can be summarized as follows:

The authors propose a new agentic framework that formalizes the process of LLM-driven heuristic generation through iterative prompting, execution, and feedback. Unlike single-turn benchmarks, HeuriGym allows LLMs to iteratively refine their algorithms by integrating structured feedback such as execution logs, verification outcomes, and cost evaluations. This closed-loop setup mirrors human-like problem-solving and provides a rigorous testbed for assessing LLM reasoning capabilities in constrained optimization settings.

They also introduce a novel set of evaluation metrics beyond traditional pass@k, most notably the Solveₛ@i metric, which measures whether an LLM can solve constrained problems within a given number of iterations across execution, solution generation, and verification stages. To assess performance quality, they define the Quality-Yield Index (QYI), combining success rate and relative solution quality to reflect both feasibility and optimality.

In addition, the paper presents a comprehensive benchmark suite of nine combinatorial optimization problems across domains such as electronic design automation, compilers, computational biology, and logistics. These problems were carefully selected to avoid overfitting to well-known tasks (like TSP or SAT) and emphasize reasoning over memorization. Each benchmark includes demonstration and evaluation sets to support few-shot and iterative learning analysis.

Finally, the authors conduct a large-scale empirical evaluation comparing several state-of-the-art LLMs—including GPT-4-mini, Gemini-2.5-Pro, Claude-3.7-Sonnet, and DeepSeek models—under the HeuriGym framework. The results show that while models like GPT-4-mini and DeepSeek-R1 achieve strong reasoning consistency, overall performance still lags behind expert-designed heuristics, highlighting the current limitations and future potential of LLMs for agentic reasoning in optimization tasks.

**Strengths:**

- **Well-presented and systematic framework:**
  The paper is clearly organized and methodologically rigorous, providing a formalized agentic framework (HeuriGym) that integrates problem definition, heuristic generation, verification, and iterative feedback in a coherent workflow.

- **Goes beyond standard benchmark tasks:**
  Unlike prior works focusing on well-studied problems such as TSP or SAT, HeuriGym introduces diverse, domain-rich combinatorial optimization tasks (e.g., operator scheduling, protein sequence design, airline crew pairing), enhancing the benchmark’s generality and realism.

- **Novel evaluation metrics capturing agentic reasoning:**
  The introduction of `Solveₛ@i` and the Quality-Yield Index (QYI) provides a more holistic evaluation than standard `pass@k`, accounting for feasibility, constraint satisfaction, and iterative refinement quality.

**Weaknesses:**

- **Lack of comparison with ILP solvers:**
  The paper claims that the proposed framework can solve larger instances than traditional ILP solvers such as Gurobi. However, no quantitative comparison is provided to substantiate this claim. For smaller-scale problems, it would be important to report the optimality gap or at least benchmark against Gurobi solutions to demonstrate relative performance.

- **Missing evaluation against state-of-the-art heuristics:**
  While the framework aims to generate heuristics automatically, it remains unclear whether the resulting solutions improve upon or even match the performance of existing state-of-the-art heuristic algorithms in the literature. Without such comparison, it is difficult to gauge whether HeuriGym advances the frontier of heuristic design or merely replicates baseline performance.

- **Marginal novelty in the agentic framework:**
  The proposed agentic framework, although well structured, shows limited conceptual innovation beyond existing approaches such as AlphaEvolve. The iterative feedback and refinement loop are largely similar in spirit, and the paper would benefit from introducing more distinctive mechanisms or theoretical insights that clearly differentiate it from prior work.

- **Limited engagement with more advanced related work:**
  Recent studies have explored more sophisticated integrations of LLMs with optimization processes. For instance, *van Stein et al. (2024)* proposed an in-the-loop hyperparameter optimization approach for LLM-based heuristic design (*ACM Transactions on Evolutionary Learning*), which achieves a tighter coupling between LLMs and evolutionary optimization. Compared with such work, HeuriGym appears relatively simplistic, relying mainly on AlphaEvolve-style iteration without deeper algorithmic co-adaptation or automated search over LLM configurations.

**Questions:**

1. In the demonstration set, only five small instances were used for each task. This sample size seems too limited to capture the variability and difficulty spectrum of real-world combinatorial problems. Could the authors clarify the rationale behind choosing such a small number of demonstration instances, and whether larger or more diverse demonstration sets were considered?

2. The paper mentions applying a citation-based filtering criterion (less than 1,000 citations) when selecting benchmark problems. This threshold appears somewhat arbitrary and potentially large, as problems with hundreds of citations could already be well-studied. What is the justification for using this particular cutoff, and how does it effectively ensure that the benchmark tasks are not overexposed or memorized by LLMs?

3. What does large solution space mean? It's not rigorously defined.

---

> ### Author Response · Authors · 2025-11-21
>
> Thanks for your insightful feedback. We address your concerns point by point below.
>
> ## Comparison with ILP solvers
> We need to mention that (1) not all the problems in our benchmark can be formulated as an ILP problem. Some problems (e.g., global routing) are hard to formulate due to complex objective structures or an intractably large number of constraints. (2) Even when an ILP formulation exists, solvers often fail to produce solutions within a reasonable time horizon because the instances are too large. For example, the intra-operation parallelism problem has a search space of $10^{65000}$ as mentioned in Section 4.2, which cannot be handled with an ILP solver.
>
> For the small subset of problems that do admit ILP formulations and remain tractable at modest instance sizes, we report the ILP results below. Specifically, three of our easiest (one-star) tasks have known optimal solutions: operator scheduling, e-graph extraction, and protein sequence design (where the provided baseline corresponds to the true biological optimum in nature). We compute the performance gap between these optima (quality as 1.0) and the LLM-generated heuristics using the Quality metric in Section 3.2 (counting only feasible LLM solutions).
>
> |  | Operator scheduling | E-graph extraction | Protein sequence design |
> |:--: |:--: |:--: |:--: |
> | Claude-3.7-Sonnet | 0.9545 | 0.5020 | 0.5360 |
> | DeepSeek-R1        | 0.9542 | 0.9363 | 0.6954 |
> | DeepSeek-V3        | 0.9410 | 0.6667 | 0.7064 |
> | GPT-o4-mini          | 0.9821 | 0.9306 | 0.6922 |
> | Gemini-2.5-Flash  | 0.9710 | 0.9282 | 0.5143 |
> | Gemini-2.5-Pro      | 0.9821 | 0.9308 | 0.4578 |
> | LLaMA-3.3-70B     | 0.4123 | 0.5019 | 0.8382 |
> | LLaMA-4-Maverick | 0.8304 | 0.0 | 0.5884 |
> | Qwen3-235B         | 0.9651 | 0.9248 | 0.7043 |
>
> The observed gaps vary across tasks. For operator scheduling, most LLM-generated solutions are within ~6% of optimal; for protein sequence design, gaps exceed 30%. Moreover, many LLM-produced solvers fail to generate any feasible solutions on several tasks, as shown by the solve@i metrics in Appendix E2. Collectively, these results demonstrate that while ILP solvers are useful where applicable, they cannot cover the full benchmark, and that current LLM-generated heuristics still have substantial room for improvement.
>
> ## Missing evaluation against SOTA heuristics
> Our work already includes such baselines. Each problem in HeuriGym is benchmarked against expert-designed heuristic solvers.
>
> As described in Section 4.1 and Section 5, for every combinatorial optimization task we provide a reproducible expert heuristic implementation drawn from the best-known domain-specific methods in the literature (e.g., operator scheduling from Cong & Zhang, 2006; LUT technology mapping using ABC). These expert heuristics are normally problem-specific heuristics instead of metaheuristics, and will serve as the reference solutions that define QYI = 1.0. Their performance is included in the evaluation via the QYI metric, and all model-generated heuristics are explicitly evaluated relative to them (Fig. 2). This ensures that LLM outputs are compared to state-of-the-art human-engineered heuristics rather than arbitrary baselines.
>
> Our results demonstrate a substantial and quantifiable performance gap. The results show that even the strongest models (e.g., GPT-o4-mini-high, Gemini-2.5-Pro) achieve weighted QYI around 0.6, meaning LLMs recover only ~60% of expert-heuristic performance on average, despite sometimes producing valid solutions. This directly answers the concern: HeuriGym reveals that current LLM-generated heuristics do not yet match or surpass state-of-the-art human-designed heuristics.

---

> ### Author Response · Authors · 2025-11-21
>
> ## Marginal novelty in the agentic framework
> We appreciate the comment and agree that many recent systems follow a similar agentic pattern. Our contribution is not a new optimizer but a benchmarking framework designed to test LLM-crafted heuristics across diverse, realistic CO domains. Unlike AlphaEvolve-style methods that evolve short programs or fill in parts of predefined metaheuristic templates, HeuriGym requires full-program synthesis from scratch, including parsing, data structures, heuristic logic, and output formatting. The agentic loop we use is deliberately simple and model-agnostic, serving only to test LLM reasoning rather than to introduce a novel search strategy. To highlight this distinction, we evaluate three state-of-the-art evolutionary frameworks (HSEvo, ReEvo, EoH) under the same iteration budget and find that all underperform our basic refinement loop on solve@10 and QYI, indicating that HeuriGym exposes limitations in existing evolutionary approaches when scaled to hundred-line solvers with complex constraints. Thus, HeuriGym should be viewed as a rigorous testbed complementary to, rather than competing with, AlphaEvolve-style systems.
>
> ## Comparison with van Stein’s work
> We thank the reviewer for highlighting this work. While LLaMEA-HPO is an advanced algorithm for tuning metaheuristics on continuous black-box problems, our aims differ: HeuriGym is a **benchmark and evaluation framework** for full-solver synthesis on CO tasks drawn from different practical domains including EDA, compilers, logistics, and computational biology. Unlike LLaMEA-HPO, which optimizes hyperparameters of compact metaheuristics, HeuriGym requires LLMs to generate complete problem-specific heuristics under fixed resource budgets. Our simple agentic loop is intentional to keep the evaluation model-agnostic and expose reasoning and repair capabilities, not to introduce a new optimizer. We view LLaMEA-HPO as complementary: it could be used on top of HeuriGym to tune prompts or agentic strategies, just as we already plug in other evolutionary frameworks (HSEvo, ReEvo, EoH), which underperform our minimal baseline on solve@10 and QYI. We will include this related work in the paper.
>
> ## Could the authors clarify the rationale behind choosing such a small number of demonstration instances, and whether larger or more diverse demonstration sets were considered?
> We chose five demonstrations per problem as a practical trade-off. As noted in the timeout settings in Appendix E1, evaluating each instance in HeuriGym is time-consuming, so substantially increasing the number of demonstrations would make the benchmark prohibitively expensive while offering limited additional benefit. A larger demo set would also risk “training on the test,” since the same demos are reused across iterations; keeping the demo set small preserves the integrity of the held-out evaluation. We also ensured that the demonstrations are diverse and capture different constraint requirements so that they can provide representative guidance. This choice aligns with few-shot LLM usage, where the goal is to assess whether models can learn from a handful of examples plus structured feedback, not from large supervised sets. Empirically (Section 5.2), even small demo sets provide meaningful gains. For example, QYI on the pickup-and-delivery problem rises from 0.2350 (0 demos) to 0.4196 (5 demos). Finally, the demonstrations are deliberately diverse within each task, offering guidance without making the benchmark easy.
>
> ## What is the justification for using this particular cutoff (1000 citations), and how does it effectively ensure that the benchmark tasks are not overexposed or memorized by LLMs?
> The 1000-citation cutoff is a practical heuristic to avoid heavily standardized “textbook” CO problems such as TSP and SAT that are almost certainly present in LLM training corpora, while still retaining well-defined, peer-reviewed tasks. In our exploratory scan, canonical benchmark problems consistently had far more than 1000 citations, whereas the domain-specific CO problems we target in EDA, compilers, logistics, and computational biology typically fell well below this threshold. We emphasize that this cutoff is not theoretically special but part of a broader anti-contamination strategy: we exclude widely benchmarked classics, use custom natural-language descriptions and I/O formats, and attach domain-specific verifiers and instance distributions that differ from any single public source. While no citation threshold can guarantee zero exposure, this combination substantially reduces the likelihood that models can rely on memorized textbook heuristics rather than genuinely synthesizing solvers for our tasks.

---

> > ### Comment · Reviewer_QPGo · 2025-11-21
> >
> > van Stein’s work is only one example of recent efforts using LLMs to discover heuristics. I should also note that the authors of this paper do not appear to fully understand that work. van Stein’s approach also used LLMs to evolve heuristics, and, in addition, incorporated hyperparameter optimization to further improve them. In this sense, the agentic framework proposed in HeuriGym seems less advanced than existing methods.
> >
> > I agree with the authors that the primary contribution of this paper is the comprehensive benchmark for evaluating LLM-generated heuristics for optimization problems. I genuinely appreciate the substantial effort involved in assembling such a benchmark. However, I do not see sufficient methodological novelty, especially given that similar works exist in the literature that employ more advanced agentic frameworks, albeit with less comprehensive benchmark instances than HeuriGym. For this reason, I believe HeuriGym may be better suited for publication in a venue such as the NeurIPS Benchmark Track.
> > I will keep my score to reflect my honest assessment (marginally below the acceptance threshold, though I do not object if the paper is ultimately accepted).

---

> > > ### Author Response · Authors · 2025-11-22
> > >
> > > Additionally, we would like to point out that “**datasets and benchmarks**” is explicitly listed as a subject area in the ICLR’26 Call for Papers (https://iclr.cc/Conferences/2026/CallForPapers) and is selected as our paper’s “Primary Area”, so the benchmark design itself is recognized as an important contribution at ICLR. We agree that van Stein’s work and many recent LLM-based heuristic discovery approaches introduce more sophisticated evolutionary strategies and prompting techniques within the agentic workflow. Our goal, however, is not to propose a new agentic workflow, but to provide a clean, reproducible baseline framework and a standard testbed into which such advanced strategies can be easily plugged, similar to the evolutionary results we presented in Table 4. These techniques are orthogonal to our contribution and can in fact use our benchmark to evaluate their improvements in a principled way. We respectfully ask the reviewer to reconsider the paper based on the value of the benchmarking problems themselves, which we believe can serve as a stronger and more comprehensive testbed for future agentic frameworks.

---

> > > > ### Comment · Reviewer_QPGo · 2025-11-22
> > > >
> > > > Thanks for pointing out the benchmark area. I was not aware of that. I've updated my score. Please have a more careful discussion of the contribution and distinction with the literature in your updated manuscript.

---

> ### Author Response · Authors · 2025-11-21
>
> ## What does a large solution space mean? It's not rigorously defined.
> For each HeuriGym task, the solution space is the set of all feasible outputs satisfying the task’s constraints, and it is “large’’ when this set grows combinatorially with problem size. For our benchmark instances, even conservative lower bounds on this space are astronomically large (e.g., $10^{65,000}$ for intra-operator parallelism), far beyond enumeration or memorization. This means each instance admits many distinct valid solutions rather than a single ground truth, and exhaustive search is infeasible. We will make this definition explicit in the revision.

---

> ### Comment · Reviewer_QPGo · 2025-11-21
>
> I would like to thank the authors for their efforts to address my comments. However, I still have several major concerns.
>
> The claim that “some problems (global routing) are hard to formulate due to complex objective structures or an intractably large number of constraints” is not accurate, because in principle any optimization problem can be formulated as a mixed-integer nonlinear program and tackled using commercial solvers such as Gurobi. I would also like to emphasize that modern solvers are far more sophisticated than plain branch-and-bound. Search spaces on the order of (10^{65000}) are routinely encountered, and much larger instances can be found in benchmark libraries such as MIPLIB.
>
> This is not intended to diminish the importance of developing LLM-generated heuristics; on the contrary, I see substantial value in this line of work. My point is simply that the specific claims about the difficulty of formulation and solver capability, as currently stated in the paper, are not correct.

---

> > ### Author Response · Authors · 2025-11-22
> >
> > We thank the reviewer for the additional feedback. Our earlier statement referred specifically to the ILP formulation, and we fully acknowledge that many optimization problems can, in principle, be expressed as mixed-integer nonlinear programs. However, in the domains we focus on, state-of-the-art tools and practical solutions are still dominated by heuristics. For example, in the VLSI global routing problem [1] included in HeuriGym, the congestion term in the objective function is highly non-linear, and there are no SOTA routers based on the MIP solvers due to the sheer problem size.
> >
> > More broadly, in engineering and scientific domains such as EDA and computer systems, commercial tools continue to rely heavily on heuristics (e.g., operator scheduling in Vitis HLS [2], technology mapping in ABC [3]). Given this landscape, we believe that exploring LLM-generated heuristics is relevant and potentially impactful.
> >
> > [1] Rongjian Liang, Anthony Agnesina, Wen-Hao Liu, and Haoxing Ren. 2024. GPU/ML-Enhanced Large Scale Global Routing Contest. In Proceedings of the 2024 International Symposium on Physical Design (ISPD '24). Association for Computing Machinery, New York, NY, USA, 269–274.
> >
> > [2] AMD Xilinx. Vitis high-level synthesis (hls), 2025. https://www.amd.com/de/products/ software/adaptive-socs-and-fpgas/vitis/vitis-hls.html.
> >
> > [3] ABC: A System for Sequential Synthesis and Verification, 2005. URL http:// www.eecs.berkeley.edu/~alanmi/abc/.

---

### Official Review · Reviewer_ZY54 · 2025-10-28

**Soundness:** 3
**Presentation:** 3
**Contribution:** 3
**Rating:** 6
**Confidence:** 5

**Summary:**

It introduces a new benchmark named HeuriGym for evaluating heuristic algorithms generated by LLMs for combinatorial optimization problems. It also presents a new criterion with three stages for the benchmark. Experiments were conducted on around 10 LLMs and different important yet not so well-known combinatorial optimization tasks. It contributes to both algorithm development and LLM communities.

**Strengths:**

LLM-driven heuristic design benchmarking is vital for both algorithm development and LLM communities, serving as an open-ended, challenging, and evaluative benchmark.

Three stages with new criteria are designed for this benchmark.

**Weaknesses:**

Further clarification is needed regarding the methodology for designing the benchmark, specifically concerning the tasks, prompts, and evaluation procedures.

As a benchmark paper, more comprehensive results, including a greater number of iterations and diverse prompt strategies, are expected.

**Questions:**

1. The problem specifications are generated by prompting DeepSeek-v3 and iteratively refined until the description is unambiguous. However, the optimal prompt for different LLMs (even different versions within the same LLM series) can vary significantly, especially for complex problem-solving tasks. While acknowledging the difficulty of designing a universal prompt, the iterative approach on a single LLM to design the prompts should be reconsidered or further clarified.

2. Please provide reasons for the timeout settings in Table 7. Why are different timeouts assigned? It appears that 10 seconds may be insufficient for many metaheuristics when implemented in Python.

3. Testing evolutionary frameworks, such as EoH and ReEvo, with only 10 iterations might be misleading. Common practice typically involves hundreds of evaluations to effectively assess evolutionary algorithms. Ten iterations with single-point-based refinement (i.e., without a population or with a population size of 1) closely resemble a random search or local refinement. To ensure effectiveness, a population size of 10 or 20 is often used. The settings for assessing evolutionary frameworks should be reconsidered, incorporating more iterations and appropriate population sizes.

4. Furthermore, I do not believe that designing a piece of an algorithm or the entire algorithm constitutes a key methodological improvement or difference in comparison with existing works, as it primarily involves letting the LLM generate longer code or complex metaheuristics. When the template and evaluation block are replaced with new settings, existing methods like EoH and ReEvo can readily be used for entire algorithm design, as evidenced in many recent works. While the benchmark's importance is not disputed, it is better to fit within the existing body of LLM-driven algorithm design research.

5. The selected tasks seem to be either too simple or too hard. For instance, results for the operator scheduling problem show 100% success for most models, even without any iteration. Conversely, why do all LLMs fail on the global routing problem (i.e., 0.0% even after 10 iterations)? Is this due to hard constraints? Would results improve with more iterations?

6. The authors argue that they focus on novel yet foundational problems, rather than commonly used combinatorial optimization tasks such as TSP and Bin Packing. However, establishing clear boundaries for task selection is challenging. Although the authors provide a methodology for selecting tasks based on Google Scholar searches, this approach can hardly guarantee the absence of pattern matching during heuristic design.

7. Could you provide the details on the training and testing instances? Are they of the same distributions and sizes? How is the out-of-distribution performance of design algorithms, which is one of the main limitations of current automated heuristic design methods.

8. A discussion and comparison with related platforms and benchmarks is suggested, such as LLM4AD and Co-Bench.
LLM4AD: A platform for algorithm design with large language model. arXiv 2024.
Co-bench: Benchmarking language model agents in algorithm search for combinatorial optimization. arXiv 2025.

From my point of view, I think the capability of LLMs to continuously improve and learn from experience is more important than zero-shot generation from prompts for algorithm/heuristic design tasks. It is similar to how human experts design high-performance algorithms through trial and error. Moreover, the results of different LLMs are not robust given different initial prompts, while the performance improvement is always observed during evolution/iterative search. However, assessing this learning capability (or, more broadly, open-ended problem-solving) in a principled manner presents a significant challenge, given its dynamic and costly evolutionary nature.

---

> ### Author Response · Authors · 2025-11-21
>
> Thanks for your insightful feedback. We address your concerns point by point below.
>
> ## The iterative approach on a single LLM to design the prompts should be reconsidered or further clarified.
> We appreciate the concern regarding prompt generalizability across various LLMs. We want to clarify that the original problem descriptions are purely drafted by humans, and the prompt used to refine the problem specifications was *not* tuned for optimal performance on DeepSeek-V3 as a solver. Instead, we leverage DeepSeek-V3 specifically as a lightweight validator to check for the soundness and clarity of the resulting specification. This process involves no complex problem-solving. Our core assumption is that if a weaker model like DeepSeek-V3 can successfully validate the specification's unambiguity and coherence, a stronger model will inherently possess sufficient understanding. Thus, the prompts are designed for general clarity, not model-specific optimization.
>
> ## Why are different timeouts assigned? It appears that 10 seconds may be insufficient for many metaheuristics when implemented in Python.
> The assignment of different timeouts is based on two distinct principles: alignment with established benchmarks and evaluation of efficient reasoning.
> 1. Alignment with original competitions. For several complex benchmarks, we adopted the established timeouts from the original competitions or literature to ensure a fair and consistent comparison:
>     * Intra-op parallelism: Timeout is set to $\mathbf{60s}$, aligning with the original contest [1].
>     * Global routing: Timeout is set to $\mathbf{300s}$, aligning with the original contest [2].
>     * Pickup and delivery w/ time windows: Timeout is set to $\mathbf{60s}$, aligning with the reported solving time in [3].
> 2. Evaluation of Efficient Reasoning (The $\mathbf{10s}$ Standard). For all remaining benchmarks, a uniform timeout of 10 seconds was intentionally selected. This is designed to evaluate a crucial aspect of the LLM's capabilities:
>     * Reasoning under Constraint: The short duration forces the model to reason about the optimal search direction given a tight time limit and real-time feedback.
>     * Heuristics vs. Metaheuristics: This constraint challenges the model to discern when simple, fast heuristics are more appropriate (e.g., for smaller instances like operator scheduling where a quick search might succeed) versus when more complex metaheuristics would lead to a timeout, thereby penalizing an inefficient approach.This two-pronged approach ensures both external validity for complex problems and an internal mechanism for evaluating time-efficient problem-solving strategies.
>
> [1] Moffitt, Michael D., and Pratik Fegade. "The asplos 2025/eurosys 2025 contest on intra-operator parallelism for distributed deep learning." Proceedings of the 30th ACM International Conference on Architectural Support for Programming Languages and Operating Systems, Volume 3. 2025.
>
> [2] Liang, Rongjian, et al. "Gpu/ml-enhanced large scale global routing contest." Proceedings of the 2024 International Symposium on Physical Design. 2024.
>
> [3] Li, Haibing, and Andrew Lim. "A metaheuristic for the pickup and delivery problem with time windows." Proceedings 13th IEEE international conference on tools with artificial intelligence. ICTAI 2001. IEEE, 2001.

---

> ### Author Response · Authors · 2025-11-21
>
> ## Testing evolutionary frameworks, such as EoH and ReEvo, with only 10 iterations might be misleading.
>
> We should clarify that we follow the standard practice used in the prior evolutionary framework papers. “The fixed budget of 10 iterations” refers to the outermost loop of the evolution (e.g., the harmony search loop in HSEvo). We set the population size to be 10 (similar setting as in Table 1 of the HSEvo paper), and in the end, each framework will generate **over 100 samples** during the process. Notice these frameworks normally incorporate multi-step reasoning for each iteration. For example, HSEvo has an internal reflection and mutation step, which will actually generate more than 10 samples for one iteration. Thus, the effective sampling budget is significantly larger than the iteration count may suggest.
>
> Classical evolutionary setups do typically run for many more generations, but those evaluations assume cheap fitness functions. HeuriGym poses a fundamentally different cost structure: each iteration requires synthesizing 300+ lines of code, compiling or interpreting the resulting program, and executing it across multiple non-trivial instances with verification and objective computation. Under these conditions, even 10 outer loops represent a **substantial computational and monetary budget** per model run. Scaling to thousands of samples is unfortunately infeasible due to the practical budget constraint.
>
> Nevertheless, we did experiment with larger population sizes of 30 and 50. However, evaluation consistently failed during the *first* iteration because the prompt design used by existing EA frameworks appends *all* sampled programs back into the context. Our programs are simply too long, causing these frameworks to exceed the LLM context window before evolution can meaningfully proceed. This limitation is mainly due to their prompt design.
>
> As we mentioned in Section 5.1, the primary reason these evolutionary frameworks underperform relative to the raw LLM baseline is that HeuriGym’s problems are far more challenging than those in their original evaluations. They involve multi-hundred-line program synthesis with intricate logical structure, and the evolutionary frameworks repeatedly fail to patch flawed initial programs. Under such circumstances, increasing the iteration count or population size does not improve performance.
>
> Our goal is not to claim that evolutionary methods are inherently weak, but to present stress-test conditions under a realistic evaluation budget. We will revise the paper to more clearly detail the experimental setup and explicitly acknowledge the limitations of applying existing evolutionary frameworks to HeuriGym.

---

> > ### Comment · Reviewer_ZY54 · 2025-11-26
> >
> > Thank you for your responses and efforts. Some of my concerns have been addressed. However, I think the statements on the evolutionary framework with LLMs might be inaccurate or misleading. For example, 1) the authors mentioned that "we follow the standard practice used in the prior evolutionary framework papers" and used about 100 evaluations for each run. However, for both EoH, ReEvo and other recent works, there are around 1,000 evaluations for each run. In HSEvo, the paper uses a fixed maximum tokens as the budget. 2) The authors mentioned that "evaluation consistently failed during the first iteration because the prompt design used by existing EA frameworks appends all sampled programs back into the context.". However, most of these methods, including EoH, ReEvo, and HSEVO, only select one or a very small number of programs into the context.

---

> > > ### Author Response · Authors · 2025-12-03
> > >
> > > We thank the reviewer for the helpful follow-up comments. We further clarify our settings and observations below:
> > > 1. Our EA configuration closely follows the setting used in HSEvo, particularly Table 1 of their paper, where the initial population is 30, later stages use a population of 10, and harmony search runs for 5 iterations, which results in less than 100 samples in total. We acknowledge that EoH and ReEvo adopt much larger evaluation budgets (on the order of ~1,000), and we agree that this can lead to better performance. However, as discussed in the paper, our setting cannot be scaled to such large budgets due to the significantly higher token cost of our full end-to-end program generation environment, which quickly becomes context- and cost-prohibitive. We believe that with more sophisticated prompt design and context engineering, these EA frameworks could benefit from larger iteration budgets; however, due to time and resource constraints, we were unable to explore higher-budget configurations.
> > > 2. We agree that not all EA frameworks append all sampled programs into the context. However, in our setting, even appending a small number of full programs is enough to exceed context limits. For example, most modern LLMs provide a context window of around 1M tokens, while a single iteration of our technology-mapping task already consumes ~158K tokens without including any prior samples or feedback. This leaves extremely limited room for incorporating additional programs. In our experiments, HSEvo often attempted to include more programs than feasible, frequently triggering context-length errors despite using its standard selection strategy.
> > >
> > > We want to emphasize that the long-program setting of HeuriGym fundamentally changes the effective context budget, thereby making standard EA prompting strategies difficult to scale without substantial redesign. We will further clarify the EA framework setting in our paper to make sure they do not convey wrong information.

---

> ### Author Response · Authors · 2025-11-21
>
> ## Designing a piece of an algorithm or the entire algorithm does not constitute a key methodological improvement or difference in comparison with existing works
> Notice that HeuriGym intentionally targets problems whose complexity demands full end-to-end algorithm construction. These tasks require models to design problem-specific data structures, reason about and enforce hard constraints, and integrate multiple stages of logic into a coherent solver. For many practical CO problems, even human experts cannot easily identify which specific component of an algorithm should be modified or improved, because there is no clear decomposition or explicit fitness function for individual pieces. This differs fundamentally from classical metaheuristic tuning, where the structure of the algorithm is fixed and only hyperparameters or small fitness functions are optimized.
>
> Our technology mapping case study (Section 5.3 and Appendix F) highlights this distinction. The current best manual heuristic relies on a multi-stage pipeline that combines dynamic programming with a Steiner-tree structure derived from LUT mapping, instead of leveraging metaheuristics. Solving this task requires synthesizing the entire pipeline; evolving a small code snippet cannot capture the intertwined logic needed to manage mapping constraints, propagate costs, and produce valid outputs.
>
> As discussed earlier, existing evolutionary frameworks further reinforce this point. Their limitations are not just performance bottlenecks but structural mismatches: they break context across iterations, repeatedly attempt to patch an already flawed seed program, and cannot incorporate the rich program-execution feedback (compiler, runtime, verification errors) provided by our environment. These failures directly stem from their design, which assumes short programs and minimal constraint handling, as shown in the results of Table 4.
> Constructing a fully adapted end-to-end evolutionary system that can evolve entire algorithms for HeuriGym-scale tasks would require non-trivial extensions such as treating runtime and verification errors as repair objectives and maintaining persistent program histories across iterations. In other words, whole-algorithm synthesis in HeuriGym is not merely a larger version of existing snippet-evolution approaches; it is a qualitatively different challenge that current frameworks are not built to handle.
>
> ## The selected tasks seem to be either too simple or too hard.
> We deliberately curated tasks spanning a wide difficulty spectrum to evaluate different model capabilities:
> 1. **High-feasibility tasks** (e.g., operator scheduling):
> A 100% yield on some tasks does not mean they are "too simple." The critical difference lies in solution quality. We use QYI, which considers both yield and quality, as the robust metric. This ensures we accurately assess performance on problems where finding a feasible solution is easy, but finding a high-quality solution is hard.
> 2. **Low-Feasibility Tasks** (e.g., global routing):
> Global routing is challenging due to an excessive number of hard constraints, making heuristic construction of feasible solutions nearly impossible. Solving it requires deep domain knowledge or knowledge transfer from similar complex constraint problems. Providing more iterations does not help generate a solution that satisfies all the constraints. The 0.0% yield highlights the current models' immediate difficulty in reasoning about these structural constraints, which is a key evaluation focus. It also reveals the shortcomings of current models.
>
> ## Although the authors provide a methodology for selecting tasks based on Google Scholar searches, this approach can hardly guarantee the absence of pattern matching during heuristic design.
> We acknowledge the reviewer's point that a Google Scholar search cannot definitively guarantee the absence of pattern matching, as establishing a clear boundary for common tasks is inherently difficult. However, the Google Scholar approach is employed as an empirical necessary heuristic proxy for ensuring task novelty. Our fundamental goal is to evaluate the model's reasoning ability, not its memorization ability. Tasks with well-established and widely published literature (e.g., SAT, TSP) inevitably come with well-established heuristics. Including these tasks risks merely evaluating whether the model has memorized these known solutions, which undermines our objective. By using the search methodology to select tasks with a less mature public knowledge base, we maximize the likelihood that the LLM is forced to reason and design novel heuristics rather than relying on cached, pre-existing patterns. This approach is a practical way to support the validity of our reasoning evaluation.

---

> ### Author Response · Authors · 2025-11-21
>
> ## Could you provide the details on the training and testing instances? Are they of the same distributions and sizes? How is the out-of-distribution performance of design algorithms?
> The training and testing instances are not drawn from the same size distribution, a decision made deliberately to evaluate the scalability and generalization of the designed algorithms.
> * Training Instances: We intentionally use smaller instances during the training loop, which allows LLMs to get feedback quickly. This choice is purely for practical efficiency to conserve computational resources and reduce the number of tokens processed.
> * Testing Instances: The test set instances are generally of a larger size.
>
> To ensure the models are not tuned to a specific scale, the prompt explicitly states that test instances will be larger and instructs the model to produce a general, scalable solution. Although this is a soft constraint, it directly probes whether the LLM can synthesize methods that generalize beyond the training distribution and encourages learning general algorithmic principles rather than instance-specific tricks. Ablation studies on few-shot demonstration in Section 5.2 and Appendix E4 actually show the effectiveness of the demo set and the ability for generalization.
>
> ## A discussion and comparison with related platforms and benchmarks is suggested, such as LLM4AD and Co-Bench.
> Thanks for pointing out the related works. We list the differences below:
> 1. Problem scopes: LLM4AD and CO-Bench focus primarily on classical, small-scale CO problems such as TSP, where once the model outputs a syntactically valid solution in the metaheuristic, feasibility is effectively guaranteed. In contrast, HeuriGym targets large, domain-specific CO tasks from EDA, compilers, logistics, and computational biology, where feasibility is nontrivial and models must discover problem-specific structure and construct tailored solvers, but not just output a scoring solution.
> 2. Scale: Existing platforms evaluate LLMs on small instance sizes; HeuriGym tests cross-size generalization by training on a smaller demo set and evaluating on significantly larger ones.
> 3. Metric design: Our agentic pipeline decomposes the pass rate into three explicit stages (execution, solution generation, and verification) and introduces QYI to capture the trade-off between solution yield and quality, offering a more diagnostic and fine-grained view of failure modes. By contrast, LLM4AD does not introduce new metrics, and CO-Bench only reports the final outcomes without stage decomposition, which can obscure important issues arising during the iterative process.
>
> We will incorporate these points into our related work section.

---

### Official Review · Reviewer_iG2L · 2025-10-29

**Soundness:** 3
**Presentation:** 3
**Contribution:** 3
**Rating:** 8
**Confidence:** 4

**Summary:**

The paper introduces HeuriGym, an agentic benchmark for testing whether large language models can invent and refine heuristic algorithms for combinatorial optimization. Instead of closed-ended Q&A, models write Python solvers, run them, see feedback, and iteratively improve under verification, mirroring real engineering workflows. Tasks span nine realistic problems across EDA, compilers, computational biology, and logistics. Performance is scored by the Quality-Yield Index (QYI)—the harmonic mean of solution quality (relative to expert heuristics) and pass rate. Across nine state-of-the-art models, top systems such as GPT-o4-mini-high and Gemini-2.5-Pro reach only ~0.6 QYI versus an expert baseline of 1.0, revealing persistent gaps in tool use, planning, and constraint handling.

**Strengths:**

- It is a well-motivated benchmark that challenges the models in their critical agentic capabilities such as tool-augmented reasoning, multi-step planning, and instruction following.
- The benchmark involves well-defined continuous objectives, large solution spaces, and agentic settings. They are suitable for benchmarking current fast-evolving LLMs.
- The paper conducts an extensive evaluation and verifies that the benchmark is challenging for SOTA models.
- The paper reveals current limitations that are insightful for future methodological development.

**Weaknesses:**

- In Table 4, you use the metric SOLVE@10. Does this mean only 10 generations are allowed for LLM+EA frameworks? This may be an unreasonable budget for EA frameworks, which typically require more iterations to achieve performance gains. Also, is it possible to incorporate feedback from your benchmark into these EA frameworks for a fairer comparison?

- Is it possible to include black-box problems, as emphasized in ReEvo, to test LLMs’ generalizable reasoning capabilities without relying on internal knowledge?

- How does this benchmark compare with other similar benchmarks, such as ALE-Bench?

**Questions:**

Please see the questions.

---

> ### Author Response · Authors · 2025-11-21
>
> Thanks for your insightful feedback. We address your concerns point by point below.
>
> ## Does SOLVE@10 mean only 10 generations are allowed for LLM+EA frameworks?
>
> We should clarify that we follow the standard practice used in the prior evolutionary framework papers. “The fixed budget of 10 iterations” refers to the outermost loop of the evolution (e.g., the harmony search loop in HSEvo). We set the population size to be 10 (similar setting as in Table 1 of the HSEvo paper), and in the end, each framework will generate **over 100 samples** during the process. Notice these frameworks normally incorporate multi-step reasoning for each iteration. For example, HSEvo has an internal reflection and mutation step, which will actually generate more than 10 samples for one iteration. Thus, the effective sampling budget is significantly larger than the iteration count may suggest.
>
> Classical EA setups do typically run for many more generations, but those evaluations assume cheap fitness functions. HeuriGym poses a fundamentally different cost structure: each iteration requires synthesizing 300+ lines of code, compiling or interpreting the resulting program, and executing it across multiple non-trivial instances with verification and objective computation. Under these conditions, even 10 outer loops represent a **substantial computational and monetary budget** per model run. Scaling to thousands of samples is unfortunately infeasible due to the practical budget constraint.
>
> Nevertheless, we did experiment with larger population sizes of 30 and 50. However, evaluation consistently failed during the *first* iteration because the prompt design used by existing EA frameworks appends *all* sampled programs back into the context. Our programs are simply too long, causing these frameworks to exceed the LLM context window before evolution can meaningfully proceed. This limitation is mainly due to their prompt design.
>
> As we mentioned in Section 5.1, the primary reason these EA frameworks underperform relative to the raw LLM baseline is that HeuriGym’s problems are far more challenging than those in their original evaluations. They involve multi-hundred-line program synthesis with intricate logical structure, and the EA frameworks repeatedly fail to patch flawed initial programs. Under such circumstances, increasing the iteration count or population size does not improve performance.
>
> Our goal is not to claim that evolutionary methods are inherently weak, but to present stress-test conditions under a realistic evaluation budget. We will revise the paper to more clearly detail the experimental setup and explicitly acknowledge the limitations of applying existing EA frameworks to HeuriGym.
>
> ## Is it possible to incorporate feedback from your benchmark into these EA frameworks for a fairer comparison?
>
> Conceptually, our environment is well-suited for feeding richer feedback into EA frameworks, and we agree this would be an interesting direction for a fairer, “best-effort” comparison. Each run of HeuriGym exposes structured diagnostic signals (syntax/runtime errors, verification failures, constraint violations), which could in principle be integrated into an EA pipeline.
>
> However, in practice, we found that incorporating such feedback into an EA pipeline is non-trivial. EA frameworks typically sample candidates from the entire pool of previously generated programs, so attaching feedback requires maintaining a large, stateful history across iterations. This design also does not scale well: many EA approaches prompt the LLM with all candidate programs, and augmenting each with feedback further inflates the prompt and quickly risks exceeding the context window.
>
> We believe that developing and tuning such feedback-aware EA+LLM pipelines constitutes a substantial research endeavor in its own right and is orthogonal to the goals of this paper.

---

> ### Author Response · Authors · 2025-11-21
>
> ## Is it possible to include black-box problems, as emphasized in ReEvo, to test LLMs’ generalizable reasoning capabilities without relying on internal knowledge?
> We appreciate this suggestion and agree that black-box optimization tasks are a valuable complement to our current benchmark. The structured problem context and explicit constraints we provide are intentional: our benchmark aims to test whether models can design valid, domain-aware heuristics, not tune an opaque function.
>
> Our primary goal is to evaluate end-to-end algorithm engineering: parsing structured inputs, managing data structures, enforcing constraints, and implementing complete solvers competitive with expert heuristics. Classical black-box optimization (as in ReEvo) mainly tests exploration-exploitation behavior, but does not assess these capabilities.
>
> Nevertheless, HeuriGym’s framework is fully compatible with true black-box problems. One can simply wrap a hidden simulator or fitness function behind our existing interface. Due to space and compute constraints, the initial release focuses on structured CO tasks with interpretable semantics and expert baselines, but we welcome the community to provide black-box problems in the future.
>
> ## How does this benchmark compare with other similar benchmarks, such as ALE-Bench?
>
> Thanks for pointing it out. ALE-Bench and HeuriGym are concurrent efforts that both evaluate iterative algorithm design, but they differ substantially in scope and evaluation philosophy. ALE-Bench focuses on AtCoder Heuristic Contest tasks, emphasizing score optimization on classic CO problems where solutions are typically based on well-known metaheuristics. In contrast, HeuriGym targets practical, high-impact domain-specific CO problems from scientific and engineering applications, each equipped with formal specifications, custom verifiers, and strong expert heuristics. Solving these tasks requires LLMs to discover *problem-specific* structure and design tailored heuristics rather than relying on generic metaheuristic templates, in order to match or surpass expert baselines. Moreover, while ALE-Bench uses ELO scores to compare against human competitors, HeuriGym provides fine-grained solve@i and QYI metrics that reveal where agents fail (execution, solution generation, or verification) and how close they are to expert solutions under fixed compute budgets. We view the two benchmarks as complementary, and we will add an explicit comparison in the related work section.

---

> > ### Comment · Reviewer_iG2L · 2025-11-26
> >
> > Thanks for your response. My concerns have been addressed.

---

### Official Review · Reviewer_NKuf · 2025-11-01

**Soundness:** 3
**Presentation:** 4
**Contribution:** 3
**Rating:** 6
**Confidence:** 4

**Summary:**

This paper introduces HeuriGen, a benchmark for evaluating LLMs on combinatorial optimization tasks that require generating and refining executable heuristics through an agentic feedback loop. Covering nine diverse problems across science and engineering, the framework measures both solution feasibility and quality using the proposed Quality–Yield Index (QYI). Results on nine leading LLMs show large performance gaps, revealing current models’ limited capabilities in adaptive reasoning and real-world problem solving.

**Strengths:**

1. I personally appreciate that this benchmark supports not only Python but also C++. Most other heuristic-generation benchmarks are limited to Python, so this cross-language support is valuable for real-world deployment scenarios and makes the benchmark more practical for diverse research settings. The released codebase is also clearly structured and of good quality, which will be appreciated by the community.

2. The problem selection process is interesting and reasonable. I was interested by the use of Google Scholar search and citation statistics as part of the selection criteria.

3. Choosing CO as the core evaluation domain for LLMs is strongly motivated. LLM for CO heuristic generation is a spot topic recently. This choice provides a solid foundation for evaluating the true problem-solving ability of LLMs in realistic, structured scenarios.

**Weaknesses:**

1. One concern is about the token usage. In the feedback loop: “After each iteration, we log the LLM-generated solution, execution trace, verification result, and evaluation score.” I feel that it appears token-inefficient and may not scale well to larger problems or longer iterations. Appendix E.9 shows multimillion-token runs, confirming substantial computational overhead. A more sustainable design might summarize or structure feedback (e.g., key errors, constraint metrics) rather than concatenating full logs, preserving the agentic realism while improving efficiency and model focus.

2. Another concern is about the fairness of the QYI design. I do like the current F-score-like design, neat and clear. But my *intuitive* feeling is that the current design puts the “quality” and “success rate” at equal importance, while in some contexts, achieving feasible solutions is more critical (at least in the beginning iterations), while in others, solution quality should dominate. Moreover, the binary definition of yield ignores near-feasible outputs and can disproportionately penalize partial success.

3. The authors mention that “we intentionally exclude ubiquitous problems”. However, I have a different opinion that this complete omission of such tasks may reduce interpretability and comparability. Classical problems serve as essential anchors, providing intuitive performance references, allowing transparent quality validation, and enabling clearer analysis of the heuristics that LLMs discover. Including even a small subset of randomized or reparameterized classical problems would not compromise contamination control and would significantly enhance the benchmark’s scientific and pedagogical value.

4. While the introduction of a weighted QYI is conceptually appealing, averaging QYI values across heterogeneous problem domains may not be statistically or conceptually justified. QYI is a relative metric that compares each problem’s ratio to its expert baseline, so cross-task comparability is limited by differences in inherent difficulty and cost scaling. For instance, achieving a 0.8 QYI on an arduous routing task may represent stronger heuristic reasoning than a 0.9 QYI on an easier scheduling task, yet the weighted average treats them equivalently.

**Questions:**

1. I have one question not directly related to the proposed contributions: in section 3.1.1, the authors mention that the optimization object and the constraints will be presented as mathematical notations. I saw some papers about LLM for heuristic generation, but they only do this by text description. Based on your knowledge and your experimental experience, do you feel that using math notations instead of pure text descriptions will bring some advantages?

---

> ### Author Response · Authors · 2025-11-21
>
> Thanks for your insightful feedback. We address your concerns point by point below.
>
> ## Feedback loop is not token-efficient and may not scale well to larger problems or longer iterations
> Our current design mirrors a human programmer’s debugging workflow: when an execution error occurs, the model receives the corresponding traceback as feedback (see Appendix E7). We agree that this prompt structure is not optimal, and techniques like summarization, structured prompting, or more compact feedback representations could further reduce tokens per iteration. However, prompt optimization is orthogonal to the primary contributions of this work. In this paper, we focus more on the benchmark construction and report baseline results using a straightforward feedback loop to establish a clear reference point. Our benchmark and framework are designed to be extensible, and we encourage users to incorporate more advanced prompting strategies to achieve improved performance.
>
> ## In some contexts, achieving feasible solutions is more critical (at least in the beginning iterations), while in others, solution quality should dominate.
> We would like to clarify that we do *not* feed in the QYI score as the metric for LLM to improve. Instead, the model always receives the **original objective value (cost)** as feedback. Therefore, it makes little difference whether QYI is high or low in the early iterations. QYI is computed only at the end of the experiment as an evaluation metric, not as part of the optimization signal.
>
> We agree that sometimes we may want to prioritize solution quality over yield, or vice versa. Our framework can accommodate such preferences. For example, we can define a weighted version of QYI analogous to the $F_\beta$ score:
>
> $$QYI_\beta:=\frac{(1+\beta^2)\cdot \text{Quality}\cdot \text{Yield}}{(\beta^2\cdot\text{Quality})+\text{Yield}}$$
>
> where $\beta$ controls the relative importance of Yield compared to Quality (with $\beta=1$ reducing to our original formulation). This parameterization allows users to flexibly adjust the trade-off between quality and yield depending on the application.
>
> ## Binary definition of yield ignores near-feasible outputs and can disproportionately penalize partial success
> While we concede that a binary yield metric may disproportionately penalize outputs that are close to feasibility, this approach is intentional and necessary for measuring utility in the final reported results. A near-feasible solution remains, by definition, non-feasible and thus not useful for downstream application.
>
> To address the need for richer feedback signals during the training phase, we include the complete output log. This supplementary information serves as the desired denser feedback mechanism, allowing the model to effectively gauge progress and refine its intermediate steps beyond the scope of the binary reward signal.
>
> ## Including even a small subset of randomized or reparameterized classical problems would not compromise contamination control and would significantly enhance the benchmark’s scientific and pedagogical value.
>
> Thank you for the suggestion. We agree that incorporating randomized or reparameterized classical problems could further enhance the benchmark. HeuriGym is designed to be an extensible platform, and after the public release, we welcome community contributions so that users can easily expand the benchmark to suit their research needs.

---

> ### Author Response · Authors · 2025-11-21
>
> ## Averaging QYI values across heterogeneous problem domains may not be statistically or conceptually justified
> We agree that aggregating QYI values across heterogeneous domains has limitations, and we do not intend the accumulated QYI to serve as a statistically rigorous comparison across fundamentally different tasks. Its purpose is primarily to provide readers with a concise, *single*-number snapshot of overall model behavior.
>
> To address this concern, we complement the aggregated metric with **comprehensive per-problem quality and yield results**, reported in Appendix E2 and E3. These finer-grained evaluations allow readers to examine model strengths and weaknesses on individual tasks and avoid any misleading conclusions that might arise from averaging across diverse domains. After the review process, all per-problem metrics and raw results will also be hosted on our benchmark website, ensuring transparency and enabling more nuanced analyses by the community.
>
> ## Using math notations instead of pure text descriptions for optimization object and the constraints will bring some advantages?
> We initially employed a hybrid approach combining text descriptions and mathematical notations for problem specifications. As mentioned in Section 4.2, through our human-in-the-loop validation process, utilizing a weak LLM to assess soundness and clarity, we empirically determined that the model exhibited a significant preference for the latter. Specifically, pure mathematical notation offers better results and unambiguity compared to natural language descriptions. Given this evidence and the requirement for a clearly defined and consistent specification for all problems, we adopted a purely mathematical specification for the objective function across the entire benchmark.

---

### Author Response · Authors · 2025-11-23

We thank the reviewers again for their thoughtful and constructive feedback. We address each concern in a separate response and have updated the draft accordingly, with all changes highlighted in red. In summary, we made the following revisions:
1. We clarified that our contribution is a benchmark (an area specifically listed as a topical interest in the ICLR’26 solicitation) and baseline agentic framework rather than an optimal pipeline integrating all advanced prompting and evolutionary techniques (Pages 2 and 7), and we expanded the discussion of limitations and future work (Page 10).
2. We clarified the experimental setup for the evolutionary frameworks and further discussed their limitations when applied to our problems (Page 8).
3. We added additional related work comparisons, including LLM4AD, ALE-Bench, CO-Bench, and LLaMEA-HPO (Page 3).
4. We updated the rationale for using Google Scholar as an empirical approach to curate the problem set (Page 6).
5. We updated the QYI formula to support different weightings of quality and yield; setting $\beta=1$ recovers the original formulation (Page 5).
6. We added discussion on mixed-integer programming and exact solvers, explaining why they are ineffective for our problem scales (Pages 6-7).
7. We clarified the selection criteria for the demonstration set (Page 6).
8. We refined the description and motivation for using a weaker LLM to validate and improve problem specifications (Page 7).

Please let us know if there are any remaining issues we should address.

---

### Author Response · Authors · 2025-12-03
**Final Remark**

We thank the reviewers again for their tremendous efforts in reviewing our papers and providing valuable feedback. We summarize the key strengths recognized by the reviewers and how we have addressed their concerns below:
1. **Well-motivated benchmark grounded in real-world CO problems.** Reviewers highlighted that HeuriGym uses practical and impactful combinatorial optimization tasks and that the selection of problems is carefully designed. (Reviewers NKuf, iG2L, QPGo)
2. **New metrics for iterative agentic evaluation.** Our solve@i and QYI metrics were noted as meaningful for capturing refinement and tool-use capability in iterative LLM workflows. (Reviewers ZY54, QPGo)
3. **Comprehensive and challenging evaluation.** The benchmark was recognized as extensively evaluated and demonstrably challenging for state-of-the-art models. (Reviewer iG2L)
4. **Insightful analysis of current limitations.** Reviewers appreciated that the paper identifies and discusses limitations that can meaningfully guide future work. (Reviewer iG2L)
5. **High-quality, reusable implementation.** HeuriGym’s well-structured and organized codebase was recognized as a strength that will facilitate adoption by the community. (Reviewers NKuf, ZY54)

The major concern that the reviewer raised is about the configuration of the evolutionary framework. We have clarified in the revised draft that we did sample more than 100 programs per run, which is consistent with the basic setting used in HSEvo. We also acknowledge that scaling to the ~1,000-sample regime used by EoH and ReEvo is currently infeasible in our setting. This is because the long-program nature of HeuriGym drastically changes the effective context budget, making standard EA prompting strategies difficult to scale without significant redesign of context management and prompt engineering.

Moreover, we also clarified that our primary contribution is a **benchmark (explicitly listed as a topic area in the ICLR’26 call for papers) together with a baseline agentic framework**, rather than an attempt to integrate all advanced prompting or evolutionary techniques into a single optimal pipeline. We have expanded the limitations and future work sections accordingly. Notably, these clarifications and the reviewer’s score increase (from 4 to 6) occurred before the information leakage, and Reviewer QPGo explicitly affirmed that “datasets and benchmarks” are a core interest area for ICLR’26 and that HeuriGym is a strong match for the venue.

In addition, we updated the manuscript to incorporate more related work and to clarify several ambiguous terms noted in the earlier comments. We do hope HeuriGym can serve as a common testbed for advancing prompting methods, evolutionary strategies, and agentic workflows for real-world program generation and combinatorial optimization.

---

### Meta-Review · Area_Chair_rEWg · 2025-12-31

**Summary:**

The paper introduces HeuriGym, an agentic benchmark for testing whether large language models can invent and refine heuristic algorithms for combinatorial optimization. HeuriGym stands out as a professionally constructed, well-motivated, and timely benchmark platform. It enables a leap in open, systematic evaluation and iteration in LLM-driven heuristic algorithm design. It will become an impactful shared resource for the field. After careful evaluation of the submission, the reviewer discussions, and the authors’ thorough point-by-point responses and manuscript revisions, I am recommending acceptance of HeuriGym to ICLR as a poster paper.

**Reviewer Concerns:**

Reviewer QPGo stated the increase in score from 4 to 6 occurred before the information leakage. Authors' rebuttal can well solve their concerns.

**Reviewer Scores:**

All reviewers provide positive ratings after rebuttal, and I don't think they have much motivation to improve their assessments further.

---

### Decision · Program_Chairs · 2026-01-26

Accept (Poster)